

# Higher-order topological superconductors from Weyl semimetals

**Ammar Jahin[1], Apoorv Tiwari[2,3] and Yuxuan Wang[1]**

**1** Department of Physics, University of Florida,
2001 Museum Rd, Gainesville, FL 32611
**2** Department of Physics, University of Zurich,
Winterthurerstrasse 190, 8057 Zurich, Switzerland
**3** Condensed Matter Theory Group, Paul Scherrer Institute,
CH-5232 Villigen PSI, Switzerland

## Abstract

We propose that doped Weyl semimetals with time-reversal and certain crystalline symmetries are natural candidates to realize higher-order topological superconductors, which exhibit a fully gapped bulk while the surface hosts robust gapless chiral hinge states. We show that in such a doped Weyl semimetal, a featureless finite-range attractive interaction favors a $p+ip$ pairing symmetry. By analyzing its topological properties, we identify such a chiral pairing state as a higher-order topological superconductor, which depending on the existence of a four-fold rotoinversion symmetry, is either intrinsic, meaning that the corresponding hinge states can only be removed by closing the bulk gap, rather than modifying the surface states, or extrinsic. We achieve this understanding via various methods recently developed for higher-order topology, including Wannier representability, Wannier spectrum, and defect classification approaches. For the four-fold rotoinversion symmetric case, we provide a complete classification of the higher-order topological superconductors. We show that such second-order topological superconductors exhibit chiral hinge modes that are robust in the absence of interaction effects but can be eliminated at the cost of introducing surface topological order.



# 1 Introduction

Topological superconductivity [1–3, 3] combines two fascinating topics in condensed matter physics, topological phases of matter and unconventional superconductivity, and is the key component of fault-tolerant topological quantum computation [4, 5]. Over the past decade, significant progress has been made in classifying topological superconductors with internal and/or crystalline symmetries. For the purpose of classification, these phases are often treated as free fermion states. For experimental realizations, much of the focus has been placed on ideas similar to the Fu-Kane superconductor [6] where a conventional superconductor is in proximity with a topological material. On the other hand, unconventional superconductors with nontrivial (i.e., non-$s$-wave) pairing symmetries can exhibit even richer symmetry-breaking and topological properties. The understanding and prediction of these unconventional topological superconductors necessarily require a synergy of band structure and electronic interaction effects.

The notion of band topology has recently been extended to *higher-order* topology [7–29], with protected gapless states localized at the corners and hinges of the sample. This opens up a new avenue for novel topological superconductivity [10, 30–36], where many interesting open questions abound, including classification of such phases and its potential application in topological quantum computation. Just like regular unconventional topological superconductors, the realization of higher-order topological superconductivity via an intrinsic pairing instability typically has stringent requirements on both the normal state band structure and the pairing symmetry in an intrinsic superconductor. There have been several recent proposals along these lines, including potential higher-order topological superconducting phases (HOTSC) in FeSeTe, in two-dimensional Dirac semimetals [10, 32, 33, 37–41], and in superconductors with unconventional $p + id$ pairing symmetry [10, 42]. Alternatively, it has been pointed out in several recent works [43, 44] that superconducting proximity effects between a quantum spin Hall insulator and a $d$-wave superconductor also realizes a HOTSC phase.

In this work we show that thanks to its normal state band structure, interacting topological semimetals with time-reversal and certain crystalline symmetries are natural candidates for hosting HOTSCs. A number of previous [45–49] works have shown that topological semimetals provide a promising avenue for realizing novel topological superconducting phases, including fully gapped ones and those with topologically protected nodal points. Here we begin by analyzing the pairing instabilities of an interacting time-reversal symmetric Weyl semimetal with four-fold rotoinversion symmetry. A minimal model of such a system consists of two bands with four co-planar Weyl points (assuming the Weyl points do not sit at the high-symmetry invariant points). We show that such systems are only consistent with $T^2 = 1$, i.e. spinless systems. With a proper chemical potential within the width of Weyl bands, there exist four Fermi pockets around each Weyl point. We find that in the presence of a finite-range attractive interaction (as opposed to an on-site or short-ranged one), the leading instability is toward a chiral $p$-wave order, which spontaneously breaks time-reversal symmetry. While the resulting superconductor is fully gapped in the bulk, it hosts gapless chiral Majorana modes at its hinges that are perpendicular to the plane of Weyl points. These gapless hinge states are characteristic of second-order topology. We examine the topological properties in the presence of a four-fold rotoinversion symmetry via several different methods, including the analysis of Wannier obstruction using symmetry indicators and find that the bulk has no well-defined Wannier representation that respects all the symmetries of the system. By examining the nature of the Wannier obstruction, we obtain its direct correspondence with the chiral hinge modes.

Using the defect classification approach that we developed for higher-order topology in an earlier work [47], we find that the defect Hamiltonian $H(\boldsymbol{k}, \theta)$ for a tube enclosing the hinge has a second Chern number protected by the four-fold rotoinversion symmetry. This further confirms the robustness of the chiral hinge modes and second-order topology.

Next, we extend our focus to the general class of four-fold rotoinversion symmetric superconductors in 3d, and obtain a full classification, which is $\mathbb{Z}_2$. In fact, our analysis can be straightforwardly extended to the case with the same symmetry and $4n$ Weyl points. As a result, the second-order topology is nontrivial when $n$ is odd. Further, we demonstrate that while the chiral hinge modes are robust for a free fermion system, they can be eliminated in the presence of strong interactions on the surface by inducing an anomalous surface topological order [29].

We also analyze the situation in the absence of a four-fold rotoinversion symmetry. Despite the reduced symmetry, the chiral $p$-wave pairing order remains the leading pairing channel. However, in the absence of four-fold rotoinversion, the aforementioned classification of HOTSC does not apply. Nevertheless, we show that the chiral hinge modes remain a robust feature of the spectrum of a finite sized sample. We show this by directly solving the defect Hamiltonian corresponding to the portion of the surface around a hinge. These hinge states can be understood as coming from *extrinsic* second-order topology, as they can be eliminated by modifying the surface without closing the gap in the bulk. The Wannier obstruction of the surface states remain present, consistent with the fact that the hinge modes are protected by the surface gap. Of important relevance to this case is a four-band Weyl semimetal with time-reversal symmetry satisfying $T'^2 = -1$. In this situation two pairs of Weyl points come from different bands that are Kramers partners, and four-fold symmetries are absent. We show that in order for chiral hinge modes to exist, addition perturbations are needed to eliminate the Kramers degeneracy while preserving a spinless version of time-reversal symmetry $T^2 = 1$.

The rest of this paper is organized as follows. In Sec. 2 we introduce the model for the normal state and analysis its pairing instabilities in the presence of an attractive interaction. In Sec. 3 we show that such a chiral $p$-wave superconductor has nontrivial second-order topology in the presence of four-fold rotoinversion symmetry. In Sec. 4 we obtain a full classification of the higher-order topology for 3d four-fold rotoinversion symmetric superconudctors, and

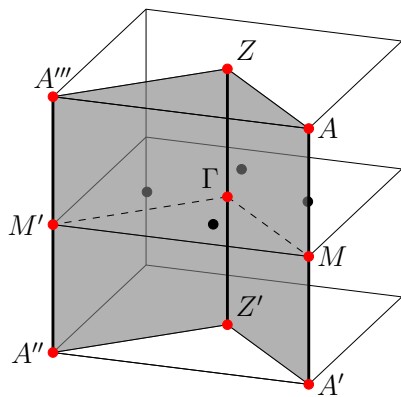

Figure 1: The full BZ with the Weyl points labeled in black dots, and the four-fold rotoinversion points labeled in red dots. Due to the $R_{4z}$ symmetry and the Weyl points, the surface $ZAA'Z'$ carry a Chern number of $1/2$.

in Sec. 5 we discuss the fate of the gapless hinge modes in the presence of strong surface interactions. In Sec. 6 we show that the chiral hinge modes remain robust in the absence of four-fold rotoinversion symmetry.

## 2 Pairing instability of a doped Weyl semimetal with time-reversal and four-fold rotoinversion symmetries

### 2.1 Normal state

We consider time-reversal ($\mathsf{T}$) symmetric Weyl semimetals, which hosts a minimum of four Weyl points. We additionally impose a four-fold rotoinversion symmetry ($\mathsf{R}_{4z}$), which is a combination of a four-fold rotation and a reflection in the $z$ direction. Note that $\mathsf{R}_{4z}$ is the simplest four-fold symmetry compatible with the alternate chirality of the four Weyl points. Without loss of generality, we conisder the configuration with one Weyl point inside each quadrant of the Brillouin zone (BZ) as shown in Fig. 1. We take the system to be gaped everywhere else.

A simple argument can be made to show that with such setup, time reversal symemtry necessarily squares to 1. i.e., $\mathsf{T}^2 = 1$. We note that the two "dividers", $AA'Z'Z$ and $A'''A''Z'Z$, and the $AA'A''A'''$ surface together encloses a Weyl point, and thus has a Chern number of one. By time reversal symmetry, the Chern number can only come from the two dividers, as the surface $AA'A''A'''$ surface is time-reversal symmetric. The two dividers are related by $\mathsf{R}_{4z}$, and thus each contributes a Berry flux $2\pi C = \pi$. Via the Stokes theorem, this Berry flux is the difference of the Berry phases (polarization) along the vertical paths $AA'$ (or $A''A'''$) and $ZZ'$. Each of the two paths are symmetric under $\mathsf{R}_{4z}$ (acting as $k_z \to -k_z$), which quantizes the polarization on these subsystems to 0 or $1/2$. For consistency with the Berry flux on the $AA'Z'Z$ (and $A'''A''Z'Z$) surface it is required that only one of the 1d subsystems to have a polarization $1/2$. Such a condition is only compatible with $\mathsf{T}^2 = 1$, since $\mathsf{T}^2 = -1$ would imply a Kramers degeneracy that necessarily leads to an integer polarization for all 1d systems. With $\mathsf{T}^2 = 1$ there are no Kramer degeneracies imposed at the time reversal invariant points. Thus we can isolate the two low-energy bands that would contribute all four Weyl points without changing the topology of the system. Consider the most general two-band lattice

model, $H = \int d\mathbf{k}\, \psi_{\mathbf{k}}^{\dagger} \mathcal{H}_n(\mathbf{k}) \psi_{\mathbf{k}}$, with the single-particle Hamiltonian given by

$$\mathcal{H}_n(\mathbf{k}) = \mathbf{f}(\mathbf{k}) \cdot \boldsymbol{\sigma} - \mu, \tag{1}$$

where $\sigma_i$'s are Pauli matrices acting on an internal band space. The Weyl nodes of the band structure are given by the condition $\mathbf{f}(\mathbf{k}_0) = 0$, which are in general isolated points in three dimensions. We impose a time-reversal symmetry $\mathsf{T}$ such that

$$\mathsf{T}\mathcal{H}_n(\mathbf{k})\mathsf{T}^{-1} = \mathcal{H}_n(-\mathbf{k}). \tag{2}$$

In general the two bands are non-degenerate other than at the Weyl points, which are not at high-symmetry points. With $\mathsf{T}^2 = 1$ we can choose the simplest form

$$\mathsf{T} = \mathcal{K}, \tag{3}$$

where $\mathcal{K}$ is the complex conjugation operator, with no loss of generality. Other choices are related by unitary transformations in the band basis. Time-reversal symmetry requires

$$f_{1,3}(-\mathbf{k}) = f_{1,3}(\mathbf{k}), \qquad f_2(-\mathbf{k}) = -f_2(\mathbf{k}). \tag{4}$$

In the presence of time-reversal symmetry, there are a minimum of four Weyl points that are pairwise related. We primarily focus on this minimal case in this work. The pair of Weyl points related by time-reversal each carry a monopole charge (Chern number) $C = 1$, while the other pair each carry $C = -1$ in accordance with the Nielson-Ninomiya theorem [50].

Under the $\mathsf{R}_{4z}$ symmetry, we have

$$\mathsf{R}_{4z}\mathcal{H}_n(\mathbf{k})\mathsf{R}_{4z}^{-1} = \mathcal{H}_n(\mathsf{R}_{4z}\mathbf{k}), \tag{5}$$

with $\mathsf{R}_{4z} : (k_x, k_y, k_z) \to (-k_y, k_x, -k_z)$. At momenta invariant under $\mathsf{R}_{4z}$, the Bloch states can be labeled by its eigenvalues. Using the fact that $f_2(\mathbf{k})$ is odd in $\mathbf{k}$, this requires that (assuming $f_{1,3}(0) \neq 0$, without loss of generality) up to a common $U(1)$ phase,

$$\mathsf{R}_{4z} \propto \exp\left[i\theta\left(\hat{f}_1(0)\sigma_x + \hat{f}_3(0)\sigma_z\right)\right], \tag{6}$$

where we defined $\hat{f}_{1,3} \equiv f_{1,3}/\sqrt{f_1^2 + f_3^2}$. (Note that we could have equivalently used the values of $\hat{f}_{1,3}$ at other high-symmetry points; self-consistency requires they all give the same results on $\mathsf{R}_{4z}$.) Further, consistency with the $f_2(\mathbf{k})\sigma_y$ term limits us to $\theta = 0$ (for which $f_2(\mathbf{k})$ needs to be even under $\mathsf{R}_{4z}$) or $\theta = \pi/2$ (for which $f_2(\mathbf{k})$ needs to be odd under $\mathsf{R}_{4z}$). A similar argument to that used to exclude the $\mathsf{T}^2 = -1$ possibility can exclude the possibility of a trivial $\mathsf{R}_{4z}$ with $\theta = 0$. In this case, we have $\mathsf{R}_{4z}^2 = 1$ and we can interpret $\mathsf{R}_{4z}$ as an inversion operation on the 1D subsystems $AA'$ and $ZZ'$. The polarization on the 1D subsystems can be determined by the inversion (played by $\mathsf{R}_{4z}$) eigenvalues at the high-symmetry points. Using known results from inversion-symmetric topological insulators [51], in terms the $\mathsf{R}_{4z}$ invariant points $\{\Gamma = (0,0,0),\ M = (\pi,\pi,0),\ Z = (0,0,\pi),\ A = (\pi,\pi,\pi)\}$, the existence of four $\mathsf{R}_{4z}$ symmetric Weyl point (i.e. the Berry flux of $\pi$ on $AA'Z'Z$ and $A'''A''Z'Z$) translates to

$$\prod_{\mathbf{k}^* \in \{\Gamma, M, Z, A\}} \eta_{\mathbf{k}^*} \equiv \eta_\Gamma \eta_M \eta_Z \eta_A = -1, \tag{7}$$

where $\eta_{\mathbf{k}^*}$ is the eigenvalue $\mathsf{R}_{4z}$ at $\mathbf{k}^*$, which takes the value of $\pm 1$ by properly choosing a common $U(1)$ phase in $\mathsf{R}_{4z}$. This clearly eliminates the possibility $\theta = 0$, for which all $\eta$'s are 1. With $\theta = \pi/2$, we have

$$\mathsf{R}_{4z} = \hat{f}_1(0)\sigma_x + \hat{f}_3(0)\sigma_z. \tag{8}$$

There are two additional composite symmetries, $C_{2z} \equiv R_{4z}^2$, and $C_{2z}T$ which generate subgroups of the full symmetry group generated by $R_{4z}$ and $T$. The symmetries act as

$$C_{2z} = -\mathbb{1}, \qquad C_{2z}T = -\mathcal{K}. \tag{9}$$

In Sec. 6 we will relax the $R_{4z}$ symmetry and only impose $C_{2z}$. From the action of the $C_{2z}T$ on the Hamiltonian it can be seen that,

$$f_{1,3}(k_x, k_y, -k_z) = f_{1,3}(k_x, k_y, k_z), \qquad f_2(k_x, k_y, -k_z) = -f_2(k_x, k_y, k_z). \tag{10}$$

The second equation implies that the Weyl points are all located at either $k_z = 0$ or $\pi$, and are therefore also related by $C_{4z}$. For concreteness, we take the 4 Weyl points to exist on the $k_z = 0$ plane with positions $\pm K$ and $\pm K'$ such that $K' = R_{4z}K$. We further focus on the low-energy fermions near the Fermi surfaces by expanding the Hamiltonian near the Weyl points,

$$h_I(\delta \mathbf{k}) \equiv \mathcal{H}_n(I + \delta \mathbf{k}) = \delta k_i \phi_I^{ij} \sigma_j - \mu, \tag{11}$$

where $I \in \{\pm K, \pm K'\}$ is the set of Weyl-point, and $\phi_I^{ij} = \partial_{k_i} f^j(\mathbf{k})\big|_{\mathbf{k}=I}$. The chirality of the Weyl points is given by $\mathrm{sgn}[\det \phi_I^{ij}]$. For later convenience, we define,

$$\epsilon_I(\delta \mathbf{k}) = \sqrt{\delta k_i [\phi_I \phi_I^T]^{ij} \delta k_j}, \tag{12}$$

$$\xi_I(\delta \mathbf{k}) = \epsilon_I(\delta \mathbf{k}) - \mu, \tag{13}$$

$$\hat{n}_I^i(\delta \mathbf{k}) = \frac{\delta k_j \phi_I^{ji}}{\epsilon_I(\delta \mathbf{k})}. \tag{14}$$

As we will make use in Sec. 2.2, we note that $\hat{\mathbf{n}}_I(\delta \mathbf{k})$ is odd in $\delta \mathbf{k}$, and $\hat{n}_I^y(\delta \mathbf{k})$ is odd under $C_{2z}T$ symmetry, i.e., under $\delta k_z \to -\delta k_z$

$$\hat{n}_I^{x,z}(\delta k_x, \delta k_y, -\delta k_z) = \hat{n}_I^{x,z}(\delta k_x, \delta k_y, \delta k_z), \tag{15}$$

$$\hat{n}_I^y(\delta k_x, \delta k_y, -\delta k_z) = -\hat{n}_I^y(\delta k_x, \delta k_y, -\delta k_z). \tag{16}$$

As a concrete example, a lattice model with $R_{4z}$ and $T$ is given by

$$f_1(\mathbf{k}) = \gamma + \cos(k_z) + \cos(k_x),$$
$$f_3(\mathbf{k}) = \gamma + \cos(k_z) + \cos(k_y),$$
$$f_2(\mathbf{k}) = \sin(k_z). \tag{17}$$

As can be easily checked, such a model has four Weyl nodes on the $k_z = 0$ plane for $-2 < \gamma < 0$. In this case

$$R_{4z} = (\sigma_x + \sigma_z)/\sqrt{2}, \tag{18}$$

and indeed the condition Eq. (7) is satisfied.

Before we move on, it is instructive to understand the orbital realization of this symmetry representation. In such a realization, time-reversal is local and $R_{4z}$ would simply swap and flip spatial coordinates, it is required that $[T, R_{4z}] = 0$, which is indeed satisfied in our case. From Eq. (8), the eigenvalues of $R_{4z}$ are $\pm 1$. Accordingly, the $|R_{4z} = 1\rangle$ degree of freedom can be realized by, e.g., a (spin-polarized) $s$-orbital or a $d_{z^2}$-orbital, while the $|R_{4z} = -1\rangle$ degree of freedom can be realized by, e.g. a $d_{xy}$-orbital or a $p_z$-orbital.

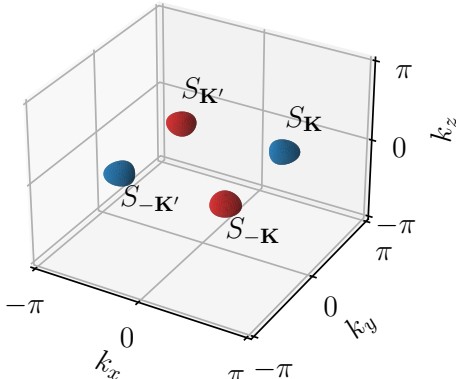

Figure 2: The position of the four ellipsoidal Fermi surfaces in the Brilliouin zone. The color of a Fermi surface denotes the chirality of the Weyl point it encloses with **red** (resp. **blue**) with $C = 1$ (resp. $-1$).

## 2.2 Analysis of the leading Cooper instability

For a finite proper chemical potential, each of the Weyl points will be surrounded with an ellipsoidal Fermi surface (FS). Let us consider the Cooper instabilities of such a WSM model in the presence of a finite-range attractive density-density interaction. The interaction is given by

$$H_{\text{int}} = - \int d\boldsymbol{k} d\boldsymbol{k}' d\boldsymbol{q} \, \psi^\dagger_{\boldsymbol{k},\alpha} \psi_{\boldsymbol{k}+\boldsymbol{q},\alpha} V(\boldsymbol{q}) \psi^\dagger_{\boldsymbol{k}'+\boldsymbol{q},\beta} \psi_{\boldsymbol{k}',\beta} \,,$$

where $\alpha, \beta$ denotes pseudospin indices, and the attractive potential depends on momentum transfer $\boldsymbol{q}$. The range of the interaction is characterized by the inverse width of the peak of $V(\boldsymbol{q})$ around $\boldsymbol{q} = 0$. For our purposes, the relevant momentum transfer are those that connect electrons on the Fermi surfaces. In the limit where $\mu$ is small, it is a good approximation to take the interaction to only depends on which of the Fermi surfaces the two electrons belong to. We define, $V_{II'} \equiv V(I - I')$, with $I, I' \in \{\pm K, \pm K'\}$ being the locations of the Weyl points as defined in Fig. 2, as the interaction between an electron on the $S_I$ Fermi surface and another on the $S_{I'}$ Fermi surface. Due to the $R_{4z}$ symmetry, we have

$$V = \begin{pmatrix} V_0 & V_1 & V_2 & V_1 \\ V_1 & V_0 & V_1 & V_2 \\ V_2 & V_1 & V_0 & V_1 \\ V_1 & V_2 & V_1 & V_0 \end{pmatrix}, \tag{19}$$

where $V_0 = V(\boldsymbol{q} = 0)$, $V_1 = V(|\boldsymbol{q}| = |\boldsymbol{K} - \boldsymbol{K}'|)$, and $V_2 = V(|\boldsymbol{q}| = 2|\boldsymbol{K}|)$.

The pairing Hamiltonian is written as,

$$H_\Delta = \int d\boldsymbol{k} \psi^\dagger_{\boldsymbol{k}} \Delta(\boldsymbol{k}) [\psi^\dagger_{-\boldsymbol{k}}]^T + \text{h.c.} \,. \tag{20a}$$

Analogous to spin-singlet and triplet pairing, one can conveniently express $\Delta(\boldsymbol{k})$ via

$$\Delta(\boldsymbol{k}) = [\Delta^0(\boldsymbol{k}) + \boldsymbol{d}(\boldsymbol{k}) \cdot \boldsymbol{\sigma}] i\sigma_y \,, \tag{20b}$$

where owing to fermion statistics $\Delta^0$ is even in $\boldsymbol{k}$ and $\boldsymbol{d}$ is odd in $\boldsymbol{k}$. Due to the lack of $SU(2)$ symmetry in the band space, the four components of the order parameter, namely $(\Delta^0(\boldsymbol{k}), d^x(\boldsymbol{k}), d^y(\boldsymbol{k}), d^z(\boldsymbol{k}))$, are in general mixed.

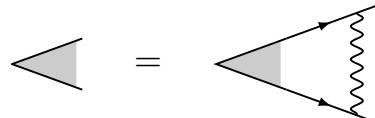

Figure 3: The linearized gap equation for the Cooper pairing vertex. See Eq. (21).

The rest of this section is devoted to the calculation of leading Cooper instability of the system, which in the presence of finite-ranged interaction we show to be $\Delta(\boldsymbol{k}) = d^y(\boldsymbol{k})\mathbb{1}$. In the weak coupling limit, the linearized gap equation, as shown by the Feynman diagram of Fig. 3, is given by

$$\Delta(\boldsymbol{k}) = T_c \sum_{\omega_m} \int d\boldsymbol{k} \ V(\boldsymbol{k} - \boldsymbol{k}') \ G(\boldsymbol{k}', \omega_m)\Delta(\boldsymbol{k}')G^T(-\boldsymbol{k}', -\omega_m), \tag{21}$$

where $\omega_m = (2m + 1)\pi T$ are the Matsubara frequencies, and the Green's function $G(\boldsymbol{k}, \omega_m) \equiv -[i\omega_m - \mathcal{H}_n(\boldsymbol{k})]^{-1}$. Using time reversal symmetry we have,

$$G^T(-\boldsymbol{k}, -\omega_m) = G(\boldsymbol{k}, -\omega_m), \tag{22}$$

which can be used to simplify the form of the gap equation. Further, the Green's functions can be approximated by projecting onto the low-energy electrons making up the FS's:

$$G_I(\delta\boldsymbol{k}, \omega_m) = -\frac{P_I(\delta\boldsymbol{k})}{i\omega_m - \xi_I(\delta\boldsymbol{k})}, \tag{23}$$

where $P_I(\delta\boldsymbol{k})$ is the projection operator onto the states near the Fermi surface,

$$P_I(\delta\boldsymbol{k}) = \frac{1}{2}\left(\mathbb{1} + \hat{\boldsymbol{n}}_I(\delta\boldsymbol{k}) \cdot \boldsymbol{\sigma}\right). \tag{24}$$

The momentum integral can be restricted to the vicinity of the four Weyl FS's, on which we assume $\Delta(\boldsymbol{k})$ takes constant values, and we have

$$\begin{aligned}\Delta_I &= T_c \sum_{\omega_m, I'} \int d\delta\boldsymbol{k} \ V_{II'}\frac{P_{I'}(\delta\boldsymbol{k})\Delta_{I'}P_{I'}(\delta\boldsymbol{k})}{\omega_m^2 + \xi_{I'}^2(\delta\boldsymbol{k})} \\ &= T_c \sum_{\omega_m, I'} \int d\delta\boldsymbol{k} \ V_{II'}P_{I'}(\delta\boldsymbol{k})\frac{\text{Tr}[\Delta_{I'}P_{I'}(\delta\boldsymbol{k})]}{\omega_m^2 + \xi_{I'}^2(\delta\boldsymbol{k})},\end{aligned} \tag{25}$$

where we define $\Delta_I = \Delta(I)$. The trace in the second line straightfowardly follows from the fact that $P_{I'}(\delta\boldsymbol{k})$ projects out one state out of two for a given $\delta\boldsymbol{k}$.

Thus, the pairing gap equation in general reduces to an eigenvalue problem for a 16 component vector (four components $(\Delta^0, \boldsymbol{d})$ for each Weyl point $I$), and strongest pairing tendency corresponds to the channel with the largest eigenvalue $T_c$. The trace in Eq. (25) can be calculated to be,

$$\text{Tr}(P_I(\delta\boldsymbol{k})\Delta_I) = i\hat{n}_I^y(\delta\boldsymbol{k})\Delta_I^0 + id_I^y - d_I^x\hat{n}_I^z(\delta\boldsymbol{k}) + d_I^z\hat{n}_I^x(\delta\boldsymbol{k}). \tag{26}$$

Additionally, using the oddness of $\hat{\boldsymbol{n}}_I(\delta\boldsymbol{k})$ under $\delta\boldsymbol{k} \to -\delta\boldsymbol{k}$ and the oddness of $\hat{n}_I^y(\delta\boldsymbol{k})$ under $\delta k_z \to -\delta k_z$ obtained previously, terms in Eq. (25) that are odd in either $\hat{\boldsymbol{n}}_I(\delta\boldsymbol{k})$ or $\hat{n}_I^y(\delta\boldsymbol{k})$ will vanish upon integrating over the solid angle,

$$\begin{aligned}\Delta_I &= i\Delta_I^0\sigma_y + id_I^y\mathbb{1} - d_I^x\sigma_z + d_I^z\sigma_x \\ &= \frac{T_c}{2}\sum_{\omega_m, I'} \int d\delta\boldsymbol{k}\frac{V_{II'}}{\omega^2 + \xi_{I'}^2(\delta\boldsymbol{k})}\Big[i\hat{n}_{I'}^y(\delta\boldsymbol{k})^2\,\Delta_{I'}^0\,\sigma_y + id_{I'}^y\mathbb{1} \\ &\quad + \left(d_{I'}^z\hat{n}_{I'}^x(\delta\boldsymbol{k}) - d_{I'}^x\hat{n}_{I'}^z(\delta\boldsymbol{k})\right)\hat{n}_{I'}^x(\delta\boldsymbol{k})\sigma_x + \left(d_{I'}^z\hat{n}_{I'}^x(\delta\boldsymbol{k}) - d_{I'}^x\hat{n}_{I'}^z(\delta\boldsymbol{k})\right)\hat{n}_{I'}^z(\delta\boldsymbol{k})\,\sigma_z\Big].\end{aligned} \tag{27}$$

We see that both the singlet, and the $d^y$ channel of the triplet pairing form independent solutions of the self-consistent equation. However the $\sigma_x$ and $\sigma_z$ channels do not, they can in general mix together

$$\Delta^0 = \frac{T_c}{2} \sum_{\omega_m, I'} \int d\delta\mathbf{k} \frac{V_{II'}}{\omega^2 + \xi_{I'}^2(\delta\mathbf{k})} \, \hat{n}_{I'}^y(\delta\mathbf{k})^2 \, \Delta_{I'}^0 \,, \tag{28}$$

$$d_I^y = \frac{T_c}{2} \sum_{\omega_m, I'} \int d\delta\mathbf{k} \frac{V_{II'}}{\omega^2 + \xi_{I'}^2(\delta\mathbf{k})} d_{I'}^y \,, \tag{29}$$

$$\begin{pmatrix} d_I^x \\ d_I^z \end{pmatrix} = \frac{T_c}{2} \sum_{\omega_m, I'} \int d\delta\mathbf{k} \frac{V_{II'}}{\omega^2 + \xi_{I'}^2(\delta\mathbf{k})} \begin{pmatrix} \hat{n}_{I'}^z(\delta\mathbf{k})^2 & -\hat{n}_{I'}^x(\delta\mathbf{k})\hat{n}_{I'}^z(\delta\mathbf{k}) \\ -\hat{n}_{I'}^x(\delta\mathbf{k})\hat{n}_{I'}^z(\delta\mathbf{k}) & \hat{n}_{I'}^x(\delta\mathbf{k})^2 \end{pmatrix} \begin{pmatrix} d_{I'}^x \\ d_{I'}^z \end{pmatrix}. \tag{30}$$

Performing the Matsubara sum we have,

$$\sum_{\omega_m} \frac{T_c}{\omega_m^2 + \xi_{I'}^2(\delta\mathbf{k})} = \frac{1}{2\xi_{I'}(\delta\mathbf{k})} \tanh \frac{\xi_{I'}(\delta\mathbf{k})}{2T_c} \,. \tag{31}$$

Doing the change of variables, $d\delta\mathbf{k} \to d\Omega\, d\xi N_{I'}(\xi, \Omega)$, and noticing that $\hat{\mathbf{n}}_{I'}(\delta\mathbf{k})$ only depend on the solid angle $\Omega$, the integral over $\xi$ reduces to,

$$\int_{-\Lambda}^{\Lambda} d\xi\, N(\xi, \Omega)\frac{1}{2\xi} \tanh \frac{\xi}{2T_c} = N(0, \Omega) \int_0^{\Lambda/T_c} dx\, \frac{1}{x} \tanh \frac{x}{2} \approx N(0, \Omega)\log\left(\frac{\Lambda}{T_c}\right), \tag{32}$$

where $\Lambda$ is an upper cutoff either from the band structure or from the interaction, and we get

$$\Delta_I^0 = \frac{1}{2}\log\left(\frac{\Lambda}{T_c^0}\right)\sum_{I'} V_{II'}\Delta_{I'}^0 \int d\Omega N_{I'}(0, \Omega)\, \hat{n}_{I'}^y(\Omega)^2 \,, \tag{33}$$

$$d_I^y = \frac{1}{2}\log\left(\frac{\Lambda}{T_c^y}\right)\sum_{I'} V_{II'}d_{I'}^y \int d\Omega N_{I'}(0, \Omega) \,, \tag{34}$$

$$\begin{pmatrix} d_I^x \\ d_I^z \end{pmatrix} = \log\left(\frac{\Lambda}{T_c^{xz}}\right)\sum_{I'} \frac{V_{II'}}{2} \int d\Omega N_{I'}(0, \Omega)\begin{pmatrix} \hat{n}_{I'}^z(\Omega)^2 & -\hat{n}_{I'}^x(\Omega)\hat{n}_{I'}^z(\Omega) \\ -\hat{n}_{I'}^x(\Omega)\hat{n}_{I'}^z(\Omega) & \hat{n}_{I'}^x(\Omega)^2 \end{pmatrix}\begin{pmatrix} d_{I'}^x \\ d_{I'}^z \end{pmatrix}. \tag{35}$$

In order to simplify the notation we make the following definitions,

$$N(0) \equiv \int d\Omega N_{I'}(0, \Omega)\,, \tag{36}$$

$$\langle \hat{n}_I^i, \hat{n}_I^j \rangle \equiv \frac{\int d\Omega N_I(0, \Omega)\hat{n}_I^i(\Omega)\hat{n}_I^j(\Omega)}{N(0)} \,. \tag{37}$$

With these notations, we obtain

$$1 = \frac{(V_0 + 2V_1 + V_2)N(0)}{2} \langle \hat{n}_I^y, \hat{n}_I^y \rangle \log\left(\frac{\Lambda}{T_c^0}\right), \tag{38}$$

$$1 = \frac{(V_0 - V_2)N(0)}{2} \log\left(\frac{\Lambda}{T_c^y}\right), \tag{39}$$

$$\begin{pmatrix} d_I^x \\ d_I^z \end{pmatrix} = \frac{(V_0 - V_2)N(0)}{2}\log\left(\frac{\Lambda}{T_c^{xz}}\right)\begin{pmatrix} \langle \hat{n}_I^z, \hat{n}_I^z \rangle & -\langle \hat{n}_I^z, \hat{n}_I^x \rangle \\ -\langle \hat{n}_I^z, \hat{n}_I^x \rangle & \langle \hat{n}_I^x, \hat{n}_I^x \rangle \end{pmatrix}\begin{pmatrix} d_I^x \\ d_I^z \end{pmatrix}, \tag{40}$$

where we have used the properties of $\Delta_0$ and $\boldsymbol{d}$ under $\mathsf{R}_{4z}$ and $\boldsymbol{k} \to -\boldsymbol{k}$. The critical temperatures can be read off the above equations as,

$$T_c^0 = \Lambda \exp\left[-\frac{2/\langle \hat{n}_I^y, \hat{n}_I^y \rangle}{(V_0 + 2V_1 + V_2)N(0)}\right], \tag{41}$$

$$T_c^y = \Lambda \exp\left[-\frac{2}{(V_0 - V_2)N(0)}\right], \tag{42}$$

$$T_c^{xz1} = \Lambda \exp\left[-\frac{2/\lambda_1}{(V_0 - V_2)N(0)}\right], \tag{43}$$

$$T_c^{xz2} = \Lambda \exp\left[-\frac{2/\lambda_2}{(V_0 - V_2)N(0)}\right], \tag{44}$$

where $\lambda_1 < \lambda_2$ are the eigenvalues of the matrix,

$$\begin{pmatrix} \langle \hat{n}_I^z, \hat{n}_I^z \rangle & -\langle \hat{n}_I^z, \hat{n}_I^x \rangle \\ -\langle \hat{n}_I^z, \hat{n}_I^x \rangle & \langle \hat{n}_I^x, \hat{n}_I^x \rangle \end{pmatrix}. \tag{45}$$

The leading instability of the system is the one that produce the highest critical temperature. We start by comparing the different triplet pairing channels together. By choice we have $T_c^{xz2} > T_c^{xz1}$. What is less trivial is comparing $T_c^y$ with $T^{xz2}$. An upper-bound on $\lambda_2$ can be obtained by replacing the off diagonal terms in Eq. (45) by their upper-bound. An upper-bound for $\langle \hat{n}_I^z, \hat{n}_I^x \rangle$ can be found using the Cauchy-Schwarz inequality,

$$\langle \hat{n}_I^x, \hat{n}_I^z \rangle \le \sqrt{\langle \hat{n}_I^x, \hat{n}_I^x \rangle \langle \hat{n}_I^z, \hat{n}_I^z \rangle}. \tag{46}$$

The charactaristic equation of the resulting matrix is,

$$\lambda \left(\lambda - \langle \hat{n}_I^x, \hat{n}_I^x \rangle - \langle \hat{n}_I^z, \hat{n}_I^z \rangle\right) = 0. \tag{47}$$

Thus finally we have,

$$\lambda_2 \le \langle \hat{n}_I^x, \hat{n}_I^x \rangle + \langle \hat{n}_I^z, \hat{n}_I^z \rangle = 1 - \langle \hat{n}_I^z, \hat{n}_I^z \rangle < 1. \tag{48}$$

Therefore we conclude that among odd-in-$\boldsymbol{k}$ solutions, $T_c^y > T_c^{xz2} > T_c^{xz1}$.

Comparing $T_c^0$ and $T_c^y$ we have two different regimes,

$$\frac{V_0 - V_2}{V_0 + 2V_1 + V_2} > \langle \hat{n}_I^y, \hat{n}_I^y \rangle, \qquad T_c^y > T_c^0, \tag{49}$$

$$\frac{V_0 - V_2}{V_0 + 2V_1 + V_2} < \langle \hat{n}_I^y, \hat{n}_I^y \rangle, \qquad T_c^y < T_c^0. \tag{50}$$

We can expect the $T_c^y > T_c^0$ when $V_0$ is dominant over $V_1$ and $V_2$, which is satisfied in the case the range of the attractive interaction is finite but sufficiently long. In this case, $d^y$ is the leading pairing order.

It is instructive to see how the calculation is carried in the special case of spherical energy contours. In this case we have,

$$[\phi_I \phi_I^T]^{ij} = v^2 \delta_{ij}, \tag{51}$$

and $N(0, \delta\boldsymbol{k})$ to be constant in $\delta\boldsymbol{k}$. We thus have,

$$N(0) = 4\pi N_I(0, \Omega), \tag{52}$$

and

$$\langle \hat{n}_I^i, \hat{n}_I^j \rangle = \frac{1}{3}\delta_{ij}. \tag{53}$$

Using this we can write,

$$
\begin{aligned}
T_c^0 &= \Lambda \exp\left[-\frac{6}{(V_0 + 2V_1 + V_2)N(0)}\right], \\
T_c^y &= \Lambda \exp\left[-\frac{2}{(V_0 - V_2)N(0)}\right], \\
T_c^{xz1} &= \Lambda \exp\left[-\frac{6}{(V_0 - V_2)N(0)}\right], \\
T_c^{xz2} &= \Lambda \exp\left[-\frac{6}{(V_0 - V_2)N(0)}\right].
\end{aligned}
\tag{54}
$$

In the spherical Fermi-surfaces case the condition for $T_c^y > T_c^0$ reduces to,

$$V_0 > V_1 + V_2. \tag{55}$$

Thus we see that in the presence of finite range interactions, it is natural to take the leading instability of the system which gaps out all the Fermi surfaces is of the $\Delta_I = d_I^y \mathbb{1}$ type.

Compatible with the Fermi statistics $\Delta_{-I}^T = -\Delta_I$, we found that such a state is an irreducible representation of $\mathsf{R}_{4z}$ that transform as

$$\mathsf{R}_{4z}\Delta_I \mathsf{R}_{4z}^T = \pm i \Delta_{\mathsf{R}_{4z}I}, \tag{56}$$

and the choice of $\pm i$ spontaneously breaks $\mathsf{T}$. This is analogous to the $p_x + ip_y$ pairing order for inversion symmetric systems. We write the pairing gap as

$$\Delta(\mathbf{k}) = (\Delta_1(\mathbf{k}) + i\Delta_2(\mathbf{k}))\mathbb{1}, \tag{57}$$

and the BdG Hamiltonian as

$$\mathcal{H}(\mathbf{k}) = \mathbf{f}(\mathbf{k})\cdot\boldsymbol{\sigma}\tau_z - \mu\tau_z + \Delta_1(\mathbf{k})\tau_x + \Delta_2(\mathbf{k})\tau_y, \tag{58}$$

where $\Delta_{1,2}(\mathbf{k})$ are real and odd in $\mathbf{k}$,

$$\Delta_{1,2}(-\mathbf{k}) = -\Delta_{1,2}(-\mathbf{k}), \tag{59}$$

and $\tau_i$ are the Pauli matrices in the Nambu space.

Even though the paring terms transform under a non-trivial one-dimension representation of four-fold rotoinversion, as given in Eq. (56), we can combine the original four-fold rotoinversion operator with a $U(1)$ charge operation under which the paring terms are invariant,

$$\tilde{\mathsf{R}}_{4z} = \mathsf{R}_{4z}e^{-i\frac{\pi}{4}\tau_z} = \left(\hat{f}_1(0)\sigma_x + \hat{f}_3(0)\sigma_z\right)e^{-i\frac{\pi}{4}\tau_z}. \tag{60}$$

It is important to emphasize that the original (physical) $\mathsf{R}_{4z}$ is broken by the $p$-wave pairing order parameter which carries a nonzero angular momentum. Since the pairing order parameter also breaks $U(1)$ symmetry in the charge sector, we have found $\tilde{\mathsf{R}}_{4z}$ as a residual composite symmetry that remains intact in the pairing state. $\tilde{\mathsf{R}}_{4z}$ should not be confused with the physical $\mathsf{R}_{4z}$; indeed we see that different from $\mathsf{R}_{4z}$, $\tilde{\mathsf{R}}_{4z}$ does not commute with $\mathsf{T}$. However, in the spatial sector, $\tilde{\mathsf{R}}_{4z}$ takes $(x, y, z) \to (y, -x, -z)$ just as $\mathsf{R}_{4z}$ does, and is equally useful for our purposes. For this reason we will still refer to $\tilde{\mathsf{R}}_{4z}$ as a four-fold rotoinversion operator.

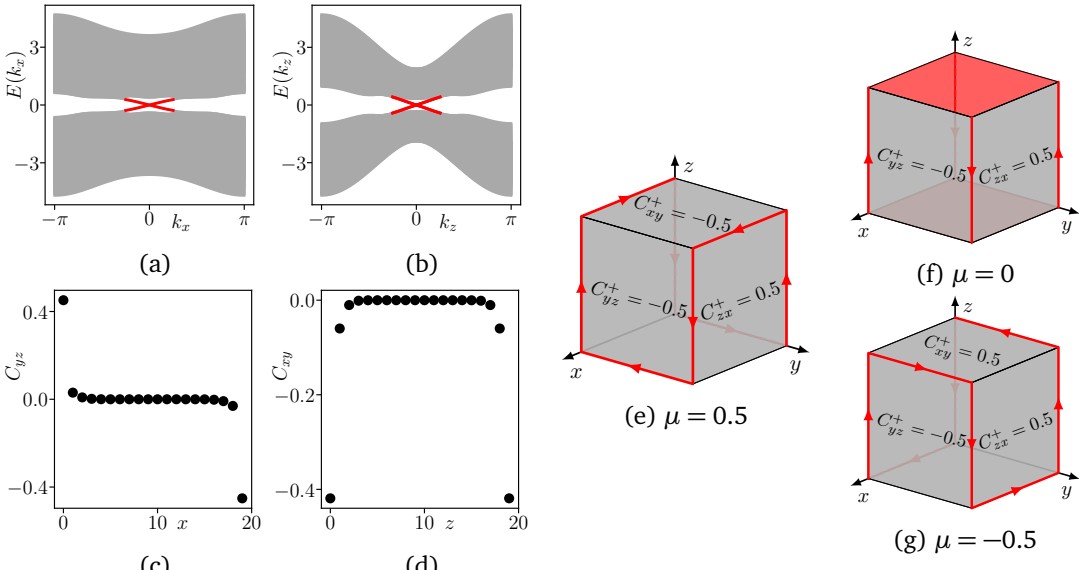

Figure 4: Panels (a-b) show the result of exact diagonalization of the Hamiltonian in Eq. (61) with $\mu > 0$ and open boundary conditions in two directions with system size $15 \times 15$ and periodic boundary condition in the third direction. Panels (c-d) shows the result of the layer resolved Chern number for the case when $\mu > 0$. Panels (e-g) shows the profile of the Majorana chiral hinge modes for different values of the chemical potential.

# 3 Higher-Order Topological superconductor with four-fold rotoinversion symmetry

In previous works [9, 10, 29], $C_{2n}T$ (with $n > 1$) symmetric HOTIs and HOTSCs have been studied and their second order topology has been analyzed in some detail. In such systems, one often finds that when defined on a $C_{2n}$ symmetric spatial geometry, the model support gapless chiral modes along hinges that are related by $C_{2n}$ symmetry. These chiral modes would intersect at points on the surface that are $C_{2n}$ invariant. This point of intersection is protected by the $C_{2n}T$ symmetry. The present situation is slightly different. Since there are no fixed points on the surface under the four-fold rotoinversion action. The symmetry does not necessitate any particular spatial position to host gapless modes. However we still find a gapless chiral mode along a four-fold rotoinversion symmetric locus on the surface that is protected by the four-fold rotoinversion symmetry. This situation is somewhat similar to the case of inversion symmetric models with second order topology [52].

We now analyze the higher-order topology of the Weyl superconductor in Eq. (58). We first numerically solve for the spectrum of a concrete tight-binding model with open boundary conditions and demonstrate the existence of chiral hinge modes. Next, by investigating the irreducible representation of the little groups of $\tilde{R}_{4z}$ at high symmetry points, we show that the system does not have a Wannier representation and is in a topological (obstructed) phase. Finally in this section we directly associate the nontrivial topology with the hinges by treating the hinges of a finite sample as defects of a space-filling system. The gapless modes hosted on the relevant hinges are naturally captured by the defect classification of topological phases.

### 3.1 Numerical Calculations of the Majorana Hinge Modes

We first present numerical results on a specific tight-binding Hamiltonian which satisfies the properties discussed in the previous section.

$$
\begin{aligned}
\mathcal{H}(\boldsymbol{k}) = {} & \left[-1 + \cos(k_z) + \cos(k_x)\right]\tau_z\sigma_x + \left[-1 + \cos(k_y) + \cos(k_z)\right]\tau_z\sigma_z - \mu\tau_z \\
& + \sin(k_z)\tau_z\sigma_y + \Delta\sin(k_x)\tau_x + \Delta\sin(k_y)\tau_y .
\end{aligned}
\tag{61}
$$

The $\tilde{\mathsf{R}}_{4z}$ symmetry for this model takes the following form,

$$
\tilde{\mathsf{R}}_{4z} = \frac{\sigma_x + \sigma_z}{\sqrt{2}} e^{-i\frac{\pi}{4}\tau_z} .
\tag{62}
$$

By taking periodic boundary condition in one direction and open boundary condition in the other two we can numerically solve for the hinge modes of the Hamiltonian in Eq. (61) using exact diagonalization. We show the results of this calculation in Fig. 4 (a-b) for the case when $\mu > 0$. Chiral modes are shown in red and we find 4 of them propagating in the $\pm k_z$ direction, and only 2 propagating in the $\pm k_{x,y}$ directions. Further checking of the localization of these chiral modes shows that indeed they are localized in the hinges, as illustrated in Fig. 4(e). We perform the same calculation but for $\mu = 0$ and $\mu < 0$. The top and the bottom surfaces are gapless for $\mu = 0$. However this gap is not protected by the $\tilde{\mathsf{R}}_{4z}$ symmetry, and depending on $\mathrm{sgn}\,\mu$, the top and bottom surfaces become gapped in different ways as shown in Fig. 4(e-g).

To better understand the topology of the system, we calculate the layer resolved Chern number on the $n$-th layer of a slab geometry defined as,

$$
C_{ij}(n) = \frac{\mathrm{Im}}{\pi} \int_{\boldsymbol{k}_\parallel} \mathrm{Tr}\left[\mathcal{P}(\boldsymbol{k}_\parallel)\partial_{k_i}\mathcal{P}(\boldsymbol{k}_\parallel)\mathcal{P}_n\partial_{k_j}\mathcal{P}(\boldsymbol{k}_\parallel)\right],
\tag{63}
$$

where $\boldsymbol{k}_\parallel = (k_i, k_j)$, are the components of the momentum parallel to the $n$-th layer, $\mathcal{P}(\boldsymbol{k}_\parallel)$ is the projection operator onto the occupied bands in the slab geometry, and $\mathcal{P}_n$ is the projection operator on the $n$-th layer. The result of this calculation for slabs parallel to the $yz$, and $xy$ planes are shown in Fig. 4(c,d). A *surface Chern number* can be defined as,

$$
C_{ij}^\pm = \sum_{n \in S^\pm} C_{ij}(n),
\tag{64}
$$

where $S^\pm$ is the set of upper/lower half of the layers. The layer resolved Chern numbers vanish for the bulk layers, hence we interpret $C_{ij}^\pm$ as a surface quantity. Restrictions imposed by $\tilde{\mathsf{R}}_{4z}$ imply,

$$
C_{xy}^+ = -C_{xy}^- , \qquad\qquad C_{yz}^+ = -C_{zx}^+ , \qquad\qquad C_{zx}^+ = -C_{yz}^- ,
\tag{65}
$$

where we take the direction of the normal vector to the surfaces to be pointing outside of the sample. Combining the above restrictions with the requirement that a chiral Majorana modes arises on the interface where this surface Chern number changes by $\pm 1$, we get that $C_{ij}^\pm$ are fixed to be either $\pm 0.5$.

For the quasi-2D slab geometry with open boundary conditions in one direction, the total Chern number can be obtained by summing over all layers and are integers as expected. In the $x$ and $y$-directions the total Chern number is zero, the total Chern number with open boundary conditions in the $z$-direction is $-\mathrm{sgn}\,\mu$, for a small $\mu$. This is despite the fact that the bulk (when periodic boundary conditions are taken in all directions) has zero Chern number on all planes in the Brillouin zone. Projecting the Majorana Chiral modes in Fig. 4(e,g), onto the $xy$-plane, one ends up with a Chiral Majorana mode circling the edges of the sample in a clockwise, or anti-clockwise fashion, consistent with the positive, or negative value of $\mu$ used in this calculation. Next, our goal is to show that the existence of the higher-order topological phase only depends on the low energy properties of the model in Eq. (58) and not on the specifics of the tight-binding model discussed here.

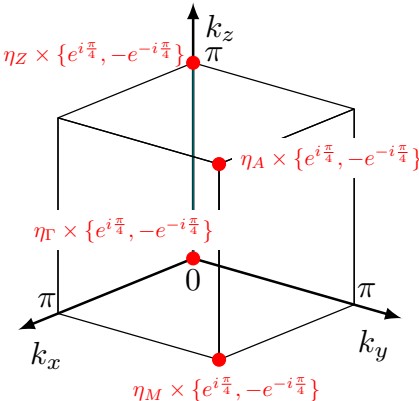

Figure 5: The eigenvalues of the $\tilde{R}_{4z}$ operator on the occupied subspace at the 4 four-fold rotoinversion symmetric points on the BZ.

## 3.2 Wannier obstruction

The pairing terms in the BdG Hamiltonian in Eq. (58) break time-reversal symmetry, thus with only $P^2 = 1$ the system is in the AZ symmetry class D. Since 3d class D systems do not support non-trivial band topology, there is no obstruction to having a well localized Wannier representation. The meaning of the Wannier representation for BdG Hamiltonian has been previously studied [30, 31, 53–55]. We therefore ask whether there exists a Wannier representation that respects the four-fold rotoinversion symmetry. We check this using a symmetry indicator approach.

If a Wannier representation exists, the centers of the Wannier functions should reproduce the eigenvalues of the symmetry operators at the high-symmetry points on the Brillouin zone. As mentioned before, there are four points in the Brillouin zone that are invariant under four-fold rotoinversion, $\{\Gamma = (0,0,0), \ M = (\pi,\pi,0), \ Z = (0,0,\pi), \ A = (\pi,\pi,\pi)\}$. All of the four-fold rotoinversion invariant points are also time-reversal invariant, and thus the pairing terms vanish and the Hamiltonian take the following form

$$\mathcal{H}(\boldsymbol{k}^*) = \boldsymbol{f}(\boldsymbol{k}^*) \cdot \boldsymbol{\sigma} \tau_z, \tag{66}$$

where $\boldsymbol{k}^* \in \{\Gamma, M, Z, A\}$, and for now we put $\mu = 0$. The eigenvalues of the four-fold rotoinversion symmetry operator, as defined in Eq. (60), on the occupied bands at these points are shown in Fig. 5.

We notice that the symmetry operators eigenvalues are completely determined by $\eta_\Gamma$, $\eta_M$, $\eta_Z$, and $\eta_A$. After a straightforward enumeration of all the possible Wannier centers and the resulting $\tilde{R}_{4z}$ eigenvalues we find the following condition for obstruction

$$\eta_\Gamma \eta_M \eta_Z \eta_A = \begin{cases} -1, & \text{obstructed}, \\ 1, & \text{not obstructed}. \end{cases} \tag{67}$$

The obstruction in the system can be understood as follows: consider a *hybrid* Wannier representation of the system that is localized in the $x$ and $y$-directions but not in the $z$-direction, $\left| v^i(R_x, R_y, k_z) \right\rangle$, $i \in \{1, 2\}$. At $k_z = 0, \pi$ the four-fold rotoinversion symmetry reduces to a fourfold rotation symmetry, $\tilde{R}_{4z} \left| v^i(R_x, R_y, k_z = 0, \pi) \right\rangle = \left| v^i(-R_y, R_x, k_z = 0, \pi) \right\rangle$. Similar 2D systems under the restriction of fourfold rotation symmetry are studied in [47]. The Wannier functions for the 2D subsystem at $k_z = 0$ $(\pi)$ are either centered at $r = (0,0)$ when $\eta_\Gamma \eta_M = 1$ $(\eta_Z \eta_A = 1)$, or at $r = (1/2, 1/2)$ when $\eta_\Gamma \eta_M = -1$ $(\eta_Z \eta_A = -1)$, where $r$ is measured relative to the unit cell center. The condition for obstruction is that only one pair, either $\eta_\Gamma$, and $\eta_M$,

or $\eta_Z$, and $\eta_A$ have a relative minus sign, but not both. The existence of the Weyl points in the $k_z = 0$ plane but not in the $k_z = \pi$ plane, ensures a relative minus sign between $\eta_\Gamma$, and $\eta_M$. Thus, in this hybrid Wannier reprsentation, the Wannier centers are centered at $r = (1/2, 1/2)$ at $k_z = 0$, and as we increase $k_z$ the Wannier centers drift and reach $r = (0,0)$ at $k_z = \pi$. This kind of *Wannier spectral flow* indicates that the system cannot be further localized in the $z$-direction. Further, following the discussion in Ref. [47], the location of the Wannier centers of $k_z = 0, \pi$ subsystems means that the 2D subsystem at $k_z = 0$ have corner Majorana zero modes while the subsystem at $k_z = \pi$ does not. This dispersion of the corner zero modes as we move from $k_z = 0$ to $k_z = \pi$ indicates the existense of the chiral hinge modes on the hinges of the 3D sample.

We note that the condition for Wannier obstruction is precisely the one in Eq. (7) we found for the existence of four Weyl points related by $R_{4z}$. Therefore, generally we have proven that an $R_{4z}$ Weyl semimetal with four Weyl nodes with attractive interaction naturally host a higher-order topological superconducting phase. This is the main result of our work.

### 3.3 Gapless hinge modes from defect classification

In this section we analyze the topology of the model in Eq. (58), from its defect classification. We treat the appearance of stable gapless states at codimension-1 or higher as a diagnostic of non-trivial bulk topology. In particular we are interested in the appearance of gapless chiral hinge modes on the four-fold rotoinversion symmetric hinges on the surface of an open geometry. To this end, consider placing the model on an open geometry that preserves the four-fold rotoinversion symmetry. Outside the sample exists a perfectly featureless atomic insulator that also preserves the spatial symmetry. As the outside region is featureless, the four Weyl-points must annihilate somewhere along the surface of the sample. Since we insist on preserving the four-fold rotoinversion symmetry, the Weyl-points are forced to annihilate at one of the four $\tilde{R}_{4z}$-symmetric points $k^* = \Gamma, M, Z$ or $A$. In any of these cases, the low energy physics is described by keeping only the leading order terms in a small momentum expansion $\delta k$ from the four-fold rotoinversion invariant point. We define,

$$
\begin{aligned}
f_{1,3}(k^* + \delta k) &= m_{1,2}, \\
f_2(k^* + \delta k) &= v_z \delta k_z, \\
\Delta_{1,2}(k^* + \delta k) &= v_x^{1,2} \delta k_x + v_y^{1,2} \delta k_y,
\end{aligned}
\tag{68}
$$

where we used the evenness of $f_{1,3}(k)$ (Eq. (4)) and the fact that $f_2(k)$ is zero over the entire $k_z = 0, \pi$ planes (Eq. (10)) from which it follows that it has no linear terms in $k_x$ and $k_y$ on these planes. Furthermore, from the odd parity nature of the pairing, and upon applying Eq. (56) twice we obtain that $\Delta_{1,2}(k)$ are even under $k_z \to -k_z$, and thus have no linear terms in $k_z$.

From the action of the four-fold rotoinversion symmetry we see that $\tilde{R}_{4z} : v^2 \to v^1$, where $v^{1,2} = (v_x^{1,2}, v_y^{1,2}, 0)$, meaning $v_x^2 = v_y^1 = v_y$, and $v_y^2 = -v_x^1 = -v_x$. The low energy continuum Weyl model in the vicinity of the four-fold rotoinversion invariant point takes the form

$$
H(\delta k) = v_{xy} \left( \delta k_x \gamma^1 + \delta k_y \gamma^2 \right) + v_z \delta k_z \gamma^3 + m_1 \gamma^4 + m_2 \gamma^5 - \mu \gamma^{12},
\tag{69}
$$

where for convenience we define $v_{xy} = \sqrt{v_x^2 + v_y^2}$, and

$$
\gamma^1 = \frac{1}{v_{xy}}(v_x \tau_x + v_y \tau_y), \quad \gamma^2 = \frac{1}{v_{xy}}(v_y \tau_x - v_x \tau_y),
$$
$$
\gamma^3 = \sigma_y \tau_z, \quad \gamma^{4,5} = \sigma_{x,z} \tau_z, \quad \gamma^{12} = i\gamma^1 \gamma^2.
\tag{70}
$$

In the bulk, the mass vector $\mathsf{m} = (\mathsf{m}_1, \mathsf{m}_2)$ is constrained such that $\mathsf{m} = \pm\mathsf{m}(\hat{f}_1(0), \hat{f}_3(0))$, with $\mathsf{m}^2 = \mathsf{m}_1^2 + \mathsf{m}_2^2$. However, it may vary as one approaches the surface. If $\mathsf{m}(\boldsymbol{r})$ represents the mass domain wall close to the surface, then $\mathsf{m}(\boldsymbol{r})$, and $\mathsf{m}(\tilde{\mathsf{R}}_{4z}\boldsymbol{r})$ are related by a reflection about the $(\hat{f}_1(0), \hat{f}_3(0))$ direction.

Below we present two complementary approaches to study the existence of hinge modes. The first approach is based on the notion of dimensional reduction/adiabatic pumping while the second approach makes use of a classification of line defects in BdG superconductors.

### 3.3.1 Via adiabatic pumping

In this section we show that the 3D class D hinge superconductor in Eq. (69) dimensionally reduces to a class BDI second-order superconductor in 2D which was studied in Ref. [47]. The four-fold rotoinversion $\tilde{\mathsf{R}}_{4z}$ reduces to a fourfold rotation $\mathsf{C}_{4z}$ in the $x$-$y$ plane. In order to make this dimensional transmutation precise, we write the low energy Hamiltonian (69) in the following suggestive way by replacing $\delta k_z \to -i\partial/\partial z$

$$H(\delta k_x, \delta k_y, z) = H_{2D}(\delta k_x, \delta k_y) + i v_z \gamma^3 \frac{\partial}{\partial z}. \tag{71}$$

We first consider setting the chemical potential $\mu = 0$. With $\mu = 0$, note that the Hamiltonian $\mathcal{H}_{2D}$ describes a class BDI superconductor. This is due to the fact that since $\{\gamma^3, \boldsymbol{H}_{2D}(\boldsymbol{k})\} = 0$, $\gamma^3$, effectively implements a chiral symmetry for the 2D model. Moreover it was shown in Ref. [47] that this model describes a BDI second-order superconductor that supports Majorana zero-modes at the corners of a $\mathsf{C}_{4z}$ symmetric spatial geometry. The states localized at each corner can be indexed by an integer $\mathsf{N}_{\mathsf{w}} \in \mathbb{Z}_{\mathrm{odd}}$ which corresponds to the difference in the number of zero-energy eigenstates with positive and negative chirality. Here we show that each such mode contributes to a chiral gapless mode on the hinge of the 3D model. Consider the ansatz of the form $|\Psi(k_x, k_y, z, t)\rangle = \phi(z, t)|\varphi(k_x, k_y)\rangle$ where $|\varphi(k_x, k_y)\rangle$ is a zero-mode of the 2D model with chirality $+1$, i.e $\boldsymbol{H}_{2D}(k_x, k_y)|\varphi(k_x, k_y)\rangle = 0$ and $\Gamma^3|\varphi(k_x, k_y)\rangle = |\varphi(k_x, k_y)\rangle$. Then solving the Schrodinger equation gives $\phi(z, t) = \phi(z + t)$. Similarly one obtains $\mathsf{N}_{\mathsf{w}}$ chiral Majorana modes with opposite chirality on adjacent corners.

The discussion above survives if we turn on a small but finite chemical potential. Indeed it was shown in Ref. [47], that that the corresponding Hamiltonian $\mathcal{H}_{2D}$ has majorana zero modes present at the corners of a $\mathsf{C}_{4z}$ symmetric spatial geometry. The topological invariant associated to these zero modes is the mod 2 reduction of the winding number $\mathsf{N}_{\mathsf{w}}$ [56]. The chirality of the hinge mode remains unchanged as compared with $\mu = 0$ case since it cannot change without a gap opening. In the next section we describe an alternate approach that provides a diagnostic for the higher-order topology based on the defect classification.

### 3.3.2 Defect invariant: Second Chern number

Let us formulate (69) as a continuum Euclidean time Dirac action

$$S = \int \mathrm{d}^3 x \mathrm{d}\tau \Psi^\dagger \left[ \partial_\tau + i \sum_{i=1}^{3} \gamma^i \partial_i + \mathsf{m}_1 \gamma^4 + \mathsf{m}_2 \gamma^5 \right] \Psi, \tag{72}$$

defined on an open spatial geometry $M$ embedded in a trivial insulator. We absorb the velocities, $v_z$, and $v_{xy}$ through an appropriate rescaling of the coordinates. Such process does not affect the topology of the system.

Comparing $\eta_{\boldsymbol{k}^*}$ in the bulk and outside $M$ i.e in the region that hosts the trivial model, they differ by a minus sign. It is known that line defects in class A and class D insulators and superconductors are integer classified and host chiral Dirac and Majorana modes respectively.

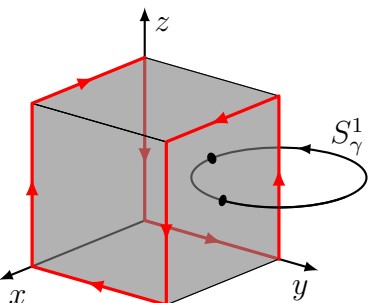

Figure 6: The choice of the $S_\gamma^1$ circumventing one of the sample hinges.

Moreover the integer invariant corresponding to a model containing a non-trivial defect is captured by the second Chern number evaluated on the hybrid four-dimensional space $\mathrm{BZ} \times S_\gamma^1$ where BZ is the 3D Brillouin zone and $S_\gamma^1$ is a real-space loop (homotopic to a circle) that links with the defect under consideration.

Such a defect invariant can directly be applied to the study of second-order topological phases in 3D by simply considering the hinge as a defect. The role of the spatial symmetries then is to ensure the stability of the defect at particular high symmetry loci on the surface of the topological phase. We consider $S_\gamma^1$ to be a path linking with a chosen hinge. For convenience we choose a path that intersects the boundary of the spatial geometry $M$ at two $\mathsf{R}_{4z}$ related points. as shown in Fig. 6. Let $\theta$ be an angular variable parameterizing the path $S_\gamma^1$. The invariant associated with the hinge, denoted as $\mathsf{N}_{\mathrm{Hinge}}$ takes the form

$$\mathsf{N}_{\mathrm{Hinge}} = \frac{1}{8\pi^2} \int_{\mathrm{BZ} \times S_\gamma^1} \mathrm{Tr}[\mathcal{F} \wedge \mathcal{F}] = \frac{1}{8\pi^2} \int_{\mathrm{BZ} \times S_\gamma^1} \mathrm{Tr}[\mathcal{P}\mathrm{d}\mathcal{P} \wedge \mathrm{d}\mathcal{P} \wedge \mathrm{d}\mathcal{P} \wedge \mathrm{d}\mathcal{P}] \,, \tag{73}$$

where $\mathcal{P} = \sum_{i=1,2} |u_i(\boldsymbol{k}, \theta)\rangle \langle u_i(\boldsymbol{k}, \theta)|$ is the projector onto the occupied states $|u_i(\boldsymbol{k}, \theta)\rangle$. In order to compute the invariant we modify our model without closing the energy gap thereby leaving the topology unaltered. More precisely, we consider the Hamiltonian

$$\widetilde{H} = \sum_{i=1}^{5} h_i(\boldsymbol{k}, \theta) \gamma^i \,, \tag{74}$$

where

$$h_i = \begin{cases} \frac{k_i - \epsilon \boldsymbol{k}^2}{\sqrt{\boldsymbol{k}^2 + \mathsf{m}^2}}, & \text{if } i = 1, 2 \,, \\ \frac{k_i}{\sqrt{\boldsymbol{k}^2 + \mathsf{m}^2}}, & \text{if } i = 3 \,, \\ \frac{\mathsf{m}_{i-3}(\theta)}{\sqrt{\boldsymbol{k}^2 + \mathsf{m}^2}}, & \text{if } i = 4, 5 \,. \end{cases} \tag{75}$$

The term $\epsilon \boldsymbol{k}^2 (\gamma^1 + \gamma^2)$ has been added as a $\tilde{\mathsf{R}}_{4z}$ symmetric regularization that implements a one point compactification of $\mathrm{BZ} \times S_\gamma^1$ such that $f$ denotes a map from $S^4$ to $S^4$. We take $\epsilon \to 0$ at the end of the calculation. Additionally, we choose a path $S_\gamma^1$ on which $\mathsf{m}^2 = \mathsf{m}_1^2 + \mathsf{m}_2^2$ is independent of $\theta$. The Hamiltonian $\widetilde{H}$ has the advantage that it is normalized with a pair of degenerate eigenstates with eigenenergies $\pm 1$. The projector onto occupied states can explicitly be written as $\mathcal{P} = \frac{1 + \boldsymbol{h} \cdot \boldsymbol{\gamma}}{2}$. Inserting this into the expression (73) one obtains

$$\mathsf{N}_{\mathrm{Hinge}} = \frac{1}{8\pi^2} \int \epsilon^{ijklm} h_i \partial_{k_x} h_j \partial_{k_y} h_k \partial_{k_z} h_l \partial_\theta h_m = \frac{1}{2\pi} \int_{S_\gamma^1} \mathsf{m} \partial_\theta \mathsf{m} \,, \tag{76}$$

therefore the topological invariant associated with a given hinge reduces to the topological winding number associated with the map $m : \theta \in S^1_\gamma \to S^1_m$ where $S^1_m$ is the circle coordinates $\arctan(m_2/m_1)$. Since (1) $\tilde{R}_{4z}$ acts as a reflection along the $(\hat{f}_1(0), \hat{f}_3(0))$ direction on the space of masses, and (2) $m$ reverses direction when moving from deep into the bulk to far outside the sample, the winding number around the loop $S^1_\gamma$ is pinned to be an odd number [47]. To conclude we have shown that the second Chern number in hybrid space $(\boldsymbol{k}, \theta)$ serves as a topological invariant which may be used to diagnose the presence of chiral Majorana hinge modes. For the Hamiltonian of the form Eq. (69) it reduces to the mass winding number around $\theta$ which is enforced to be non-vanishing and odd by the spatial $\tilde{R}_{4z}$ symmetry. Therefore we emphasize that although the tube does not preserve $\tilde{R}_{4z}$ symmetry, the second Chern number is indeed quantized is an odd integer due to this spatial symmetry.

# 4  Classification of $\tilde{R}_{4z}$-symmetric higher-order superconductors

In this section we derive the classification of $\tilde{R}_{4z}$-symmetric higher-order phases. We treat the appearance of robust ingappable modes on high symmetry lines and points on the surface of a fully gapped and spatially symmetric superconductor as diagnostics of second and third order topology respectively. There have been several recent works on the classification of higher-order topology in BdG superconductors [30, 31, 53–55]. These works focus on extending the framework of symmetry indicator [57, 58] as diagnostics for band insulators to the context of mean-field descriptions of superconductors. For the purpose of classification, it is convenient to work with ground states directly rather than with Hamiltonians [59–66]. A ground state of a model within a certain topological phase with a given crystalline symmetry $G$ can be adiabatically deformed to a particular type of state known as *block state*. A block state corresponding to a higher-order topological phase can be understood hueristically as a network of lower dimensional topological states with only internal symmetries glued together in a manner that is compatible with all spatial symmetries.

Here we illustrate the construction for the case of $\tilde{R}_{4z}$-symmetric class D superconductors. To do so, we consider a cell complex that admits an action of four-fold rotoinversion, illustrated in Fig. 7. The cell complex consists of a collection of $p$-cells. Topologically, an $p$-cell is equivalent to a $p$-dimensional disc without its boundary. Since we are interested in higher-order topology and therefore boundary modes, we consider the cellulation of an open four-fold rotoinversion symmetric geometry. The cell complex consists of a network of 1-cells and 2-cells. Note that we do not consider 3-cell as (i) they do not affect the classification of higher order phases and (ii) for the present case, i.e class D, there are no topologically non-trivial phases in 3D. Moreover, we also do not consider bulk 0-cells since they do not contribute to any boundary signatures. We consider a cell complex such that each $p$-cell is either left entirely invariant or mapped to another $p$-cell under the under the action of four-fold rotoinversion. Since, the four-fold rotoinversion only has a single fixed-point, and we do not consider 0-cells, all the $p$-cells we consider transform to four-fold rotoinversion related $p$-cells under the symmetry action. It is therefore convenient to divide up the $p$-cells into four-fold rotoinversion orbits. There are 3 bulk and 4 boundary 2-cell orbits which in Fig. 7, we denote as $\alpha, \beta, \gamma$ and $a, \ldots, d$ respectively. Likewise there are 2 bulk and 9 boundary 1-cell orbits which we denote as $\Lambda_{1,2}$ and $A, \ldots, G$ respectively.

A particular bulk state is constructed by populating a chosen orbit or more generally a collection of orbits by non-trivial topological states with the constraint that the bulk be fully gapped for the chosen network. More concretely, since class D superconductors in 1D and 2D are $\mathbb{Z}_2$ and $\mathbb{Z}$ classified respectively with the 1D Kitaev chain and the 2D $p \pm ip$ superconductors as generators, we may populate the bulk of the $\tilde{R}_{4z}$-cellulation with states corresponding to

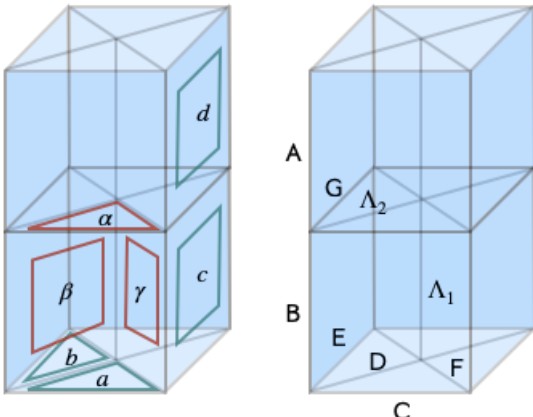

Figure 7: An illustration for a choice of cell complex for the point group $\tilde{R}_{4z}$ on an open geometry. The 2-cells are illustrated in panel (a) wherein the 2-cells $a, b, c, d$ are each a representative of a certain $\tilde{R}_{4z}$ orbit on the surface of the depicted geometry. Likewise $\alpha, \beta$ and $\gamma$ each label a certain $\tilde{R}_{4z}$ orbit in the bulk of the geometry. Similarly, panel (b) illustrates the distinct $\tilde{R}_{4z}$ orbits for the 1-cells. The representatives of the surface orbits are denoted $A, B, \dots, G$ while the bulk orbit representatives are denoted as $\Lambda_1$ and $\Lambda_2$.

the $p \pm ip$ and Kitaev phase on some combination of the $\alpha, \beta, \gamma$ and $\Lambda_{1,2}$ orbits respectively. Let the state assigned to the $\alpha$-orbit have topological index $n_\alpha \in \mathbb{Z}$ and similarly for $\beta$ and $\gamma$, likewise we denote the index assigned to the 1-cells belonging to the orbits $\Lambda_{1,2}$ as $m_{1,2}$. A priori bulk states are therefore labelled by $(n_\alpha, n_\beta, n_\gamma, m_1, m_2) \in \mathbb{Z}^3 \times \mathbb{Z}_2^2$. Since each of these candidate bulk cells contribute gapless 1D modes or zero modes on the boundaries of the cells, we must ensure that these modes can be gapped out pairwise such that one ends up with a fully gapped bulk. Notably we require $n_\beta + n_\gamma = 0$ such that the central hinge ($\Lambda_1$) is gapped. Upon imposing this condition, the bulk is fully gapped, since (i) the gapless modes contributed by the $\gamma$ and $\beta$ orbits on the 1-cells $\Lambda_2$, cancel out pairwise upon imposing the condition $n_\beta + n_\gamma = 0$ and (ii) the gapless modes contributed by the $\alpha$ orbit cancel out pairwise. Therefore the most general fully gapped bulk state is labelled as $(n_\alpha, n_\beta, -n_\beta, m_1, m_2) \in \mathbb{Z}^2 \times \mathbb{Z}_2^2$. Each non-trivial bulk cell contributes a gapless mode on the boundary such that one ends up with a network of gapless currents and zero-modes on the boundary as illustrated in Fig. 8.

Next, we ask which of the above modes are truly the signature of bulk topology. To answer this question, one needs to check which modes can be annihilated or equivalently constructed from a purely surface pasting of $p \pm ip$ and Kitaev states. Firstly, it can be checked that the $m_2$ Majorana modes constributed on the surface by the presence of Kitaev state on $\Lambda_2$ can be trivialized by surface pasting of Kitaev chains on the orbits corresponding to the 1-cells A and F. Similarly one can transform the configuration $(n_\alpha, n_\beta, -n_\beta, \dots)$ to $(n_\alpha - n_\beta, 0, 0, \dots)$ by surface pasting of $n_\beta$ copies of $p + ip$ states on the $a$ and $d$ orbits. Collectively, these two operations reduce the space of non-trivial bulk states from $\mathbb{Z}^2 \times \mathbb{Z}_2^2$ to $\mathbb{Z} \times \mathbb{Z}_2$ indexed by $(n_\alpha - n_\beta, 0, 0, m_1, 0)$. It can be verified that the $m_1$ zero modes contributed by $\Lambda_1$ are robust, hence there exist a $\mathbb{Z}_2$ classified third order superconductor protected by point group $R_{4z}$. Getting back to the $n_\alpha - n_\beta$ chiral majorana mode propagating around the sample on the reflection symmetric plane. One can always change $n_\alpha$ to $n_\alpha + 2n$ by pasting $n$ copies of $p \pm ip$ states on all the surface orbits $a, b, c, d$. This reduces the classification of second-order phases to $\mathbb{Z}_2$. To summarize the classification of both second and third order $\tilde{R}_{4z}$ symmetric superconductors in class D is $\mathbb{Z}_2$. For second order superconductors, this is generated by the bulk state with the $\alpha$-orbit populated with $p + ip$ class D superconductors while for the third order topology, it is generated by the

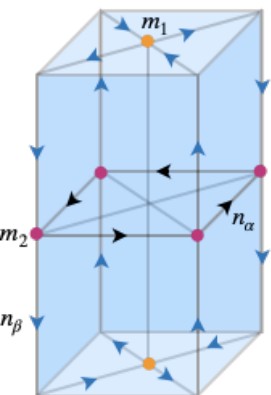

Figure 8: An illustration of a state with a fully gapped bulk a surface containing an $\tilde{R}_{4z}$ symmetric configuration of chiral majorana modes and majorana zero-modes. The hinges with blue and black arrows contain $n_\alpha$ and $n_\gamma$ majorana modes while the orange and red dots denote the presence of $m_1$ and $m_2$ majorana zero modes.

populating the $\Lambda_1$ orbit with Kitaev chains. Furthermore, we note that the above classification can be readily extended to the case of higher-order topological insulators protected by $R_{4z}$, by simply replacing the 2D block states with Chern insulator states rather than $p + ip$ states. Upon doing so, one recovers a $\mathbb{Z}_2$ second-order topology for $R_{4z}$ symmetric insulators which agrees with the known classification for non-interacting second-order insulators protected by four-fold rotoinversion symmetry. The diagnostic of such a topology is the appearance of an ingappable chiral Dirac mode on an four-fold rotoinversion symmetric hinge.

We end this section with a discussion on how our results extends to situations with $4n$, $n \in \{0, 1, 2, \cdots\}$ Weyl points, while still preserving $T^2 = 1$, and $R_{4z}$ symmetries. There are two ways the model can be modified to include more Weyl points. The first is by adding copies of the minimal model already studied to each other. Since our classification of the system is $\mathbb{Z}_2$, the chiral superconductor obtained this way is topological when $n$ is odd, and trivial when $n$ is even. The second way to modify the model is to change the dispersion of the two bands such that they host more than 4 Weyl points. In this case it is straightforward to follow our analysis of the symmetry indicators and defect approach that the existence of an even (odd) number of Weyl points per BZ quadrant, regardless of their relative charge, would lead to a trivial (topological) bulk invariant. Thus, we reach the conclusion that, in general, systems with $4n$ Weyl points are topological $n$ is odd.

# 5 $\tilde{R}_{4z}$ symmetric second-order superconductor with surface topological order

In previous sections, we showed that class D superconductors enriched by rotoreflection symmetry supports non-trivial second order topology. The appearance of a robust chiral majorana hinge mode on a rotoreflection symmetric line on the surface was treated as diagnostic of the second-order topology. Here we ask whether these surface modes remain robust in the presence of symmetry preserving strong interactions on the surface. We answer this question in the negative by constructing a fully gapped topologically ordered surface that preserves all the symmetries in question. We construct such a surface topological order (STO) by symmetrically introducing $SO(3)_6$ non-abelian topological orders on the two $\tilde{R}_{4z}$ related regions denoted $\Sigma_{1,2}$ in Fig. 9. A similar construction for the topologically ordered surfaces of $C_{2n}\mathcal{T}$-symmetric second-order topological superconductors has been previously studied in [29]. The

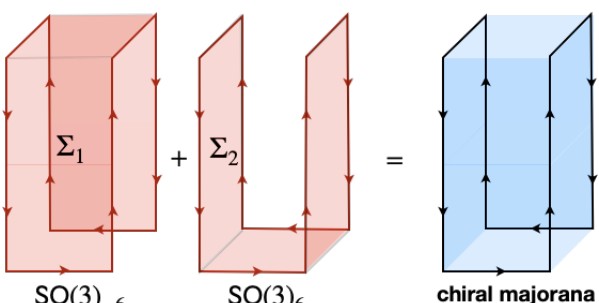

Figure 9: The chiral Majorana hinge mode on the surface of an $\tilde{R}_{4z}$-symmetric second-order superconductor can be gapped out by introducing a topologically-ordered surface. The figure illustrates an $\tilde{R}_{4z}$-symmetric pattern of $SO(3)_6$ topological order which furnishes a single chiral Majorana hinge mode that can gap out the hinge mode contributed from the bulk second-order superconductor.

$SO(3)_6$ topological order is a 'spin' or fermionic topological order [67] as it contains a single fermionic excitation (denoted below as $j = 3$) which is local, in the sense that it braids trivially with all other excitations/anyons in the topological order. Such a model is described by the continuum Chern-Simons action [68, 69]

$$S_I = \frac{(-1)^I k}{4\pi} \int_{M_I} \text{Tr} \left\{ A \wedge dA + \frac{2}{3} A \wedge A \wedge A \right\}, \tag{77}$$

where $k$ is the 'level' of the Chern-Simons theory which is 6 for present purpose, $A$ is $SO(3)$-valued gauge connection and $M_I = \Sigma_I \times S^1$ with $I = 1, 2$ labelling the two $\tilde{R}_{4z}$-related regions and $S^1$ is the compactified time domain. The $SO(3)_6$ topological order has a total of four anyons labelled $j = 0, 1, 2, 3$, with $j = 3$ being a fermion [29, 70, 71] and $j = 0$ the vacuum sector or "trivial anyon". The $j = 1, 2$ anyons are semionic and anti-semionic respectively. The fusion rules among the anyons are

$$j \times j' = \sum_{j''=|j-j'|}^{\min(j+j', 6-j-j')} j'', \tag{78}$$

while the modular $S$ and $T$ matrices that describe the braiding and self-statistics respectively are given by

$$T_{j,j'} = \exp\{2\pi i j(j+1)/8\} \delta_{j,j'}, \qquad S_{j,j'} = \frac{1}{2} \sin\left[\frac{(2j+1)(2j'+1)\pi}{8}\right]. \tag{79}$$

Since the regions $\Sigma_1$ and $\Sigma_2$ share a common hinge as their boundary, one obtains two sets of co-propogating chiral edge modes on the hinge, one from each of the surface topological orders. Each of these correspond to a chiral $SO(3)_6$ Wess-Zumino-Witten (WZW) conformal field theory (CFT) [72] with chiral central charge $c_- = 9/4$. The combined CFT on the hinge has a central charge $c_- = 9/2$. We denote the holomorphic current operators as $\mathcal{J}_{a,I}$ where $I = 1, 2$ again labels which topological order the mode is contributed from and $a = 1, \ldots, \dim(\mathfrak{so}(3))$. The current operators satisfy the operator product expansion

$$\mathcal{J}_{a,I}(z) \mathcal{J}_{b,I}(w) \sim \frac{k \delta^{ab}}{(z-w)^2} + \frac{i f_{ab}^c \mathcal{J}_{c,I}}{z-w}, \tag{80}$$

where $f_{abc}$ are the structure constants of the $\mathfrak{so}(3)$ Lie-algebra. The Hamiltonian of the hinge CFT is obtained via the Sugawara construction [73] and takes the form

$$H_0 = \frac{1}{k + h^\vee} \sum_{I,a} \mathcal{J}_{a,I} \mathcal{J}_{a,I}. \tag{81}$$

The modes of the current operators additionally satisfy the Kac-Moody algebra that acts on the states in the conformal field theory, which are thus organized into conformal towers or representations of the Kac-Moody algebra. Each representation is built on a highest weight state which is related to a conformal primary operator via the state operator map and is in one-to-one correspondence with the bulk anyons. We label the primary operators just as the bulk anyons by a tuple $(j_1, j_2)$ where $j_I = 0, 1, 2, 3$. One obtains conformal characters $\chi_{j_1, j_2}$ by tracing over the corresponding conformal towers $\mathcal{H}_{(j_1, j_2)}$

$$\chi_{(j_1, j_2)}(\tau) = \text{Tr}_{\mathcal{H}_{(j_1, j_2)}}\left[ e^{2\pi i \tau (H_0 - \frac{c}{24})} \right], \tag{82}$$

where $H_0$ is the Hamiltonian in Eq. (81) and $\tau$ is the modular parameter of the spacetime torus $\partial M_I$. The bulk topological data in Eq. (79) can be recovered from the edge CFT by performing the S (i.e $\tau \to -1/\tau$) and T (i.e. $\tau \to \tau + 1$) modular transformations on the conformal characters. Next, we deform the Hamiltonian in Eq. (81) by adding terms that lead to a condensation on the hinge. Such a condensation is equivalent to adding 'simple currents' to Kac-Moody algebra which furnishes a so-called *extended chiral algebra*. The simple currents that can be simultaneously condensed correspond to primary operators that are mutually local (i.e have a trivial S-matrix element) and have integer spin (i.e have a trivial T matrix element). Adding simple currents to the chiral algebra further constrains the corresponding representation theory and therefore has profound physical consequences on the structure of the theory. Some of the conformal towers merge together while others are removed from the spectrum. In the present case, there are three candidate simple current operators corresponding to the primaries $(j_1, j_2) = (1, 2), (2, 1)$ and $(3, 3)$. These primaries correspond to the only 'condensable' operators as they exhaust all the integer spin operators in the theory. We denote this set as $\mathcal{B}$ and add the following term to the Hamiltonian in Eq. (81)

$$H = H_0 + \lambda \sum_{(j_1, j_2) \in \mathcal{B}} (\Phi_{(j_1, j_2)} + \Phi^{\dagger}_{(j_1, j_2)}). \tag{83}$$

At strong coupling i.e. $\lambda \to \infty$, this leads to a theory with a single non-trivial representation corresponding to a chiral majorana fermion with $c_- = 9/2$. More precisely, the sectors $(0, 0), (1, 2), (2, 1), (3, 3)$ form the new vacuum of the theory while the sectors $(1, 1), (2, 2), (0, 3), (3, 0)$ are identified into a single fermionic sector. The remaining sectors get confined. The $c_- = 9/2$ mode can be mapped to single chiral Majorana mode with $c_- = 1/2$ by symmetric surface pasting of $p + ip$ superconductors described in Sec. 4. Therefore by inducing topological order on the surface, it is possible to assemble a pattern of chiral currents that corresponds to the hinge modes obtained from a non-trivial $\tilde{R}_{4z}$ symmetric second-order superconductor. As a corollary one can completely gap out the surface of second-order $\tilde{R}_{4z}$ symmetric superconductor by inducing surface topological order.

# 6 Boundary-Obstructed Topology with twofold rotation symmetry $C_{2z}$

In this section we study the case where the spatial four-fold rotoinversion symmetry is broken down to the $C_{2z}$ subgroup. We find that a BdG model with four (modulo eight) Weyl-points and $C_{2z}$ symmetry still furnishes a topological superconductor which supports a chiral Majorana hinge mode on its surface. However the mode is no longer protected by the bulk topology and instead is *boundary-obstructed*, in the sense that it can be gapped out by a purely surface deformation.

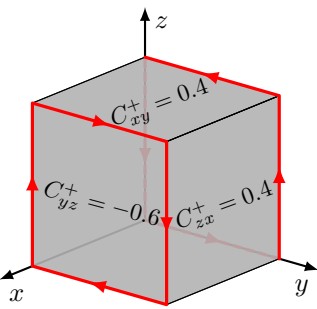

Figure 10: The Majorana zero modes of the model Hamiltonian in Eq. (84). The surface Chern numbers $C_{ij}^{\pm}$ are as defined in Eq. (64).

## 6.1 Boundary-obstruction and Wannier representation

Before discussing the topology of our system with symmetry broken down to $\mathsf{C}_{2z}$, we briefly discuss how this symmetry reduction affects the Cooper instability of the system. We still expect the normal state to have the Weyl points on the $k_z = 0, \pi$ planes since they were pinned on the planes by $\mathsf{C}_{2z}\mathsf{T}$ symmetry. Additionally, we still expect a minimum of 4 Weyl points, a pair at $\pm K$ and another at $\pm K'$. Even though the two pairs are not related by any symmetry of the system, we cannot have only a single pair due to the fact that each Weyl-point in a pair related by time-reversal symmetry have the same chirality. This, in conjunction with the Nielsen Ninomiya theorem requires a minimum of two pairs.

In the absence of the $\tilde{\mathsf{R}}_{4z}$ symmetry, one no longer requires $|\Delta_K| = |\Delta_{K'}|$. This however does not change the fact that $\Delta_I = d_I^y \mathbb{1}$ still is an eigenmode of the self-consistent equation. Moreover, we still expect a regime in which it is the leading instability as it remains to be the only mode that completely gaps out the Fermi-surfaces of the Weyl semimetal.

We illustrate boundary-obstructed topology in the $\mathsf{C}_{2z}$-symmetric case via a specific simplified model,

$$H(\boldsymbol{k}) = \left[\gamma_x + \cos(k_x)\right]\sigma_x\tau_z + \sin(k_z)\sigma_y\tau_z + \left[\cos(k_y) + \cos(k_z) - 1\right]\sigma_z\tau_z - \mu\tau_z$$
$$+ \sin(k_y)\tau_x + \sin(k_x)\tau_y. \tag{84}$$

Numerically solving for the chiral Majorana hinge modes, we obtain the profile shown in Fig. 10. The sample has two separate chiral modes that are related by $\mathsf{C}_{2z}$ symmetry. These Majorana chiral modes can be removed by for example gluing two 2D $p + ip$ superconductors with opposite Chern numbers on the two opposite $xz$-surfaces without breaking the symmetry. The model can therefore at best be boundary-obstructed. From the point of view of bulk Wannier representability, the case with only $\mathsf{C}_{2z}$ symmetry is simpler than the case with the more restrictive $\tilde{\mathsf{R}}_{4z}$ symmetry. The only restriction of $\mathsf{C}_{2z}$ is for the Wannier centers to come in pairs that are related by the symmetry, but otherwise the exact positions can be arbitrary. This might seems counter-intuitive at first, since the existence of the chiral modes on the hinges indicate the existence of some sort of a Wannier obstruction. If the bulk is Wannier representable, the only remaining possibility is that the stand-alone surface not be Wannier representable. We discuss this in some detail. The terms in model in Eq. (84) can be re-organized as

$$H(\boldsymbol{k}) = H_{p+ip}(\boldsymbol{k}) + H_{\text{SSH}}(\boldsymbol{k}), \tag{85}$$

with

$$H_{p+ip}(\boldsymbol{k}) = \left[\cos(k_y) + \cos(k_z) - 1\right]\sigma_z\tau_z + \sin(k_z)\sigma_y\tau_z + \sin(k_y)\tau_x,$$
$$H_{\text{SSH}}(\boldsymbol{k}) = \left[\gamma_x + \cos(k_x)\right]\sigma_x\tau_z + \sin(k_x)\tau_y. \tag{86}$$

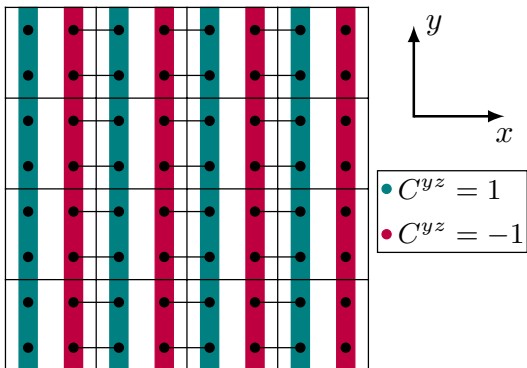

Figure 11: The model in Eq. (84) can be viewed as the stacking of Chern super-conducting layers with SSH like coupling between the layers. In the fully dimerized limit, it is clear that the bulk of the system is Wannier representable, whereas the surfaces perpendicular to the $x$-direction are not.

The $\mathcal{H}_{p+ip}(\boldsymbol{k})$ term describes two 2-dimensional layers parallel to the $yz$-plane with opposite Chern numbers trivially stacked, while the $\mathcal{H}_{\text{SSH}}(\boldsymbol{k})$ term describes an SSH-like coupling between the layers as shown in Fig. 11. An insulating (i.e without particle-hole symmetry) version of this model is also discussed in [74]. Looking at the case when $\gamma_x = 0$, as in Fig. 11, it is clear that the surfaces of the sample (when cut in the $yz$-plane) are not Wannier representable because of the dangling $p + ip$ superconducting layer at each end. Away from the $\gamma_x = 0$ limit the situation is less obvious. However, the Wannier states would evolve smoothly as we move away from the fully dimerized limit, thus the situation would remain unchanged.

## 6.2 Defect approach

We show that the low energy properties of the general Hamiltonian in Eq. (58) even in the absence of the $\tilde{\text{R}}_{4z}$ symmetry leads to a surface theory that is gapped in a topologically non-trivial way, leading to hinge chiral modes. We consider the system with cylindrical hinges along the $z$-directions. We take the radius of the cylinder to be much larger than the inter-atomic distance. The surface theory at each point on the surface of the cylinder can then be taken as that of a straight edge tangent to that point. The rounded hinge can be parametrized by an angle $\theta$ and we define $\hat{\boldsymbol{n}}_\perp(\theta)$ as the unit vector perpendicular to the tangent surface, and $\hat{\boldsymbol{n}}_\parallel(\theta)$ as the direction parallel to the surface and the $xy$-plane. Thus at each point on the surface, $\hat{\boldsymbol{n}}_\perp(\theta)$, $\hat{\boldsymbol{n}}_\parallel(\theta)$, and $\hat{\boldsymbol{n}}_z$) constitute and orthonormal coordinate basis. See Fig. 12 for an ilustration of the geometry.

Since we are interested in the low energetics of the system, we study the system near the Weyl points, and take the order parameter to be small of order $\epsilon$ and write,

$$\Delta_{1,2}(\boldsymbol{k}) = \epsilon g_{1,2}(\boldsymbol{k}). \tag{87}$$

If we start with a particle near the $\boldsymbol{K}$ point, a surface in the $\theta$ direction would *scatter* the particle back, flipping its momentum in the $\hat{\boldsymbol{n}}_\perp(\theta)$ direction. Generically, the momentum of this scattered particle will not coincide with another Weyl point. A special case is when $\hat{\boldsymbol{n}}_\perp(\theta)$ is in the same direction as $\boldsymbol{K}$, in which the surface mix the momenta at the $\boldsymbol{K}$ point with the $-\boldsymbol{K}$ point. We label such special direction with $\theta_0$. We will reserve the subscripts $\parallel, \perp$, and $z$ to indicate the components in the $\hat{\boldsymbol{n}}_\parallel(\theta_0)$, $\hat{\boldsymbol{n}}_\perp(\theta_0)$, $\hat{\boldsymbol{n}}_z$ respectively.

We expand the Hamiltonian near the Weyl points for a small momentum deviation $\boldsymbol{q}$, and introduce a valley degree of freedom, $\nu_z$, such that $\nu_z = 1$ (respectively $-1$) indicate the $\boldsymbol{K}$

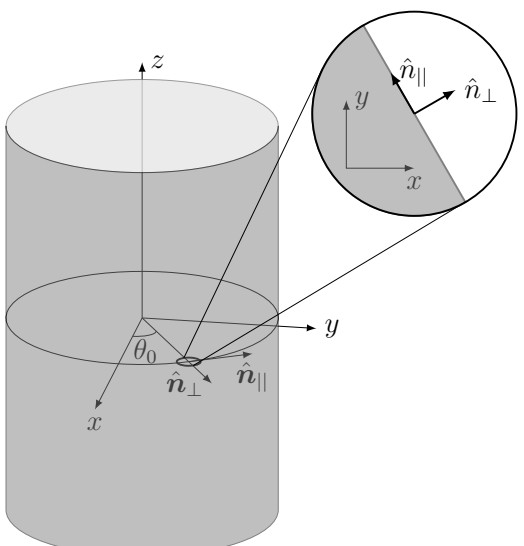

Figure 12: Real space geometry of the sample.

(respectively $-\mathbf{K}$) point. We define, $g_i \equiv g_i(\mathbf{k})\big|_{\mathbf{k}=\mathbf{K}}$ and

$$\vec{\phi}_i \equiv \frac{\partial f_i(\mathbf{k})}{\partial \vec{k}}\bigg|_{\mathbf{k}=\mathbf{K}}, \quad \vec{\gamma}_i \equiv \frac{\partial g_i(\mathbf{k})}{\partial \vec{k}}\bigg|_{\mathbf{k}=\mathbf{K}}, \tag{88}$$

and set $q_{||} = q_z = 0$, keeping only the first order terms in $\epsilon$ and $q_\perp$, and let $q_\perp \to -i\partial_\perp$. The resulting Hamiltonian can be written as,

$$H_0 = -i(\phi_{1\perp}\sigma_x + \phi_{3\perp}\sigma_z)\tau_z \nu_z \partial_\perp + \epsilon(g_1\tau_x + g_2\tau_y)\nu_z. \tag{89}$$

Note that $\phi_{2\perp} = 0$ since from Eq. (10) $f_2(\mathbf{k})$ is zero over the entire $k_z = 0$ plane where the Weyl points are located. We solve this equation on the half-infinite plane with the vacuum on the $r_\perp > 0$ side. This equation has the following zero modes solutions,

$$\psi^\alpha(r_\perp) = \chi^\alpha e^{\Delta_0 r_\perp / \nu_\perp}, \tag{90}$$

where we define,

$$\begin{aligned} \nu_\perp &= \sqrt{\phi_{1\perp}^2 + \phi_{3\perp}^2}, \\ \Delta_0 &= \epsilon\sqrt{g_1^2 + g_2^2}, \end{aligned} \tag{91}$$

and $\chi^\alpha$ is a eight-component spinor (coming from two band, two valleys, and two Nambu sectors) determined by the following condtions. First, for the zero mode solution to hold, we have

$$\tilde{\sigma}_x \tilde{\tau}_y \chi^\alpha = +\chi^\alpha, \tag{92}$$

with

$$\tilde{\sigma}_x \equiv \frac{1}{\nu_\perp}(\phi_{1\perp}\sigma_x + \phi_{3\perp}\sigma_z), \qquad \tilde{\sigma}_y \equiv \sigma_y, \qquad \tilde{\sigma}_z \equiv i\sigma_y\tilde{\sigma}_x, \tag{93}$$

$$\tilde{\tau}_x \equiv \frac{\epsilon}{\Delta_0}(g_1\tau_x + g_2\tau_y), \qquad \tilde{\tau}_z \equiv \tau_z, \qquad \tilde{\tau}_y \equiv i\tau_z\tilde{\tau}_x. \tag{94}$$

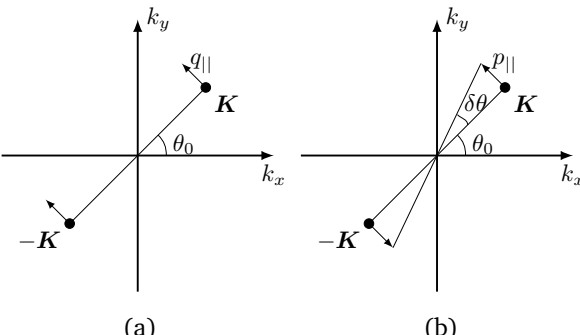

Figure 13: The relative change in momentum between the two valleys for (a) adding momentum to the particles to find the dispersion, (b) changing the direction of the surface by a $\delta\theta$.

Second, the boundary mode is a superposition between incoming and outgoing waves with $\pm K$, or $\nu_z = \pm 1$, depending on the detailed form of the boundary potential. Without loss of generality, in the valley basis, we choose the condition set by the boundary potential to be

$$\nu_x \chi^\alpha = -\chi^\alpha. \tag{95}$$

This is equivalent to the boundary condition used in Ref. [75]. There exist two such eight-component spinors satisfying the above boundary conditions. Next we find the form of the boundary Hamiltonian for a generic $q_{||}$ and $q_z$ and for a generic angular position $\theta = \theta_0 + \delta\theta$ on the surface. For a fixed angular position, the deviation in momenta at the $K$, and $-K$ points has the same direction, see Fig. 13(a). Upon projecting to the two-dimensional subspace for the boundary states, we get

$$h(q_{||}, q_z) = \hat{P}\left[\phi_{2z}\tilde{\sigma}_y\tilde{\tau}_z q_z + \beta q_{||}\tilde{\tau}_y\right], \tag{96}$$

where $\hat{P}$ is the projection onto the subspace and we have defined,

$$\beta = \frac{\epsilon}{2}\operatorname{Tr}\tilde{\tau}_y(\gamma_{1||}\tau_y + \gamma_{2||}\tau_x). \tag{97}$$

For a different surface parameterized by the angle $\theta = \theta_0 + \delta\theta$, the axis of $q_{||} = 0$ is rotated by $\delta\theta$. In the new coordinate system, effectively the perturbation incurred are opposite shifts $p_{||} = \pm|K|\delta\theta$ in the positions of Weyl points, shown in Fig. 13(b). It turns out that the perturbation terms that survives projection onto the two-dimensional subspace is

$$h(\delta\theta) = \hat{P}m\tilde{\sigma}_z\tilde{\tau}_z\delta\theta, \tag{98}$$

where

$$m = \frac{|K|}{2}\operatorname{Tr}\tilde{\sigma}_z(\phi_{1||}\sigma_x + \phi_{3||}\sigma_z). \tag{99}$$

Putting the two perturbations together we get a two-band Hamiltonian,

$$h(q_{||}, q_z, \delta\theta) = \hat{P}\left[\phi_{3z}q_z\tilde{\sigma}_y\tilde{\tau}_z + \beta q_{||}\tilde{\tau}_y + m\delta\theta\tilde{\sigma}_z\tilde{\tau}_z\right], \tag{100}$$

which describes a 2D Dirac fermion with a mass domain wall at $\delta\theta = 0$. Such a Hamiltonian is known to host chiral propagating modes that are localized at the domain wall [76,77]. This concludes our proof.

### 6.3 Two-band vs four-band Weyl semimetals

So far we have restricted out discussion on *two-band* Weyl semimetals – that is, the four Weyl points are formed by two bands across the full Brillouin zone, which are non-degenerate except at Weyl points. Since there are no Kramers degeneracy at high-symmetry points, necessarily the time-reversal symmetry satisfies $T^2 = 1$.

In Sec. 2 we have remarked that the spin-full version of time-reversal symmetry with $T^2 = -1$ is incompatible with $\tilde{R}_{4z}$ symmetry. However, it is possible to retain only a two-fold rotational symmetry $C_{2z} = \tilde{R}^2_{4z}$ and have $T^2 = -1$. Due to the additional Kramer's degeneracy, such a Weyl semimetal involves four bands, given by the following Hamiltonian $H = \int d\boldsymbol{k}\, \psi^\dagger_{\boldsymbol{k}} \mathcal{H}_n \psi_k$ where

$$H_n(\boldsymbol{k}) = f_1(\boldsymbol{k})\sigma_x + f_2(\boldsymbol{k})\sigma_y + f_3(\boldsymbol{k})\sigma_z + f'_3(\boldsymbol{k})\sigma_z s_x - \mu\,, \tag{101}$$

where $s_z$ is the Pauli matrix representing an additional spin degree of freedom, $f_{1,3}(\boldsymbol{k})$ are even functions and $f'_3(\boldsymbol{k})$ and $f_2(\boldsymbol{k})$ are odd. Such a Hamiltonian preserves a time-reveral symmetry $T' = is_y K$ that squares to $-1$. The two-fold rotation symmetry is represented as $C_{2z} = is_z$. The location of the Weyl points are given by the conditions

$$f_1(\boldsymbol{k}) = 0,\ f_2(\boldsymbol{k}) = 0,\ f_3(\boldsymbol{k}) = \pm f'_3(\boldsymbol{k})\,. \tag{102}$$

As a concrete example, such a Weyl semimetal with four Weyl points ais realized by the lattice model in which

$$\begin{aligned} f_1(\boldsymbol{k}) &= \cos k_x + \cos k_y + \cos k_z - 2\,,\quad f_3(\boldsymbol{k}) = 1/2\,, \\ f'_3(\boldsymbol{k}) &= \sin k_x\,,\quad f_2(\boldsymbol{k}) = \sin k_z\,. \end{aligned} \tag{103}$$

It is straightforward to show that a $p + ip$ pairing order, e.g., with

$$\int d\boldsymbol{k}\, \psi^\dagger_{\boldsymbol{k}} [\Delta_x \sin(k_x) + i\Delta_y \sin(k_y)] \sigma_z s_z (\psi^\dagger_{\boldsymbol{k}})^T + h.c. \tag{104}$$

gaps out all Fermi surfaces enclosing the Weyl points. However, one can readily verify that such a system does *not* host chiral hinge modes, even though the low-energy spectrum *in the bulk* is identical to that of the two-band model. It turns out that the low-energy surface states, which we relied on in the previous subsection to derive the hinge states, in general are *not* solely determined by the low-energy bulk states. In particular, having a four-band normal state, the boundary conditions given by Eq. (92) and (95) does not reduce the boundary modes to a two-dimensional subspace.

This obstacle can be removed by lifting the $T'$ symmetry. This removes all the Kramers degeneracies at high symmetry points and one can separate the four-band model into one with two Weyl bands and two remote bands. For example, one can include a perturbation from a $T'$ breaking, $C_{2z}$ preserving term $\sim M s_z \sigma_z$. As long as $M$ is sufficiently small, it does not affect the band structure near the Weyl points, but it lifts the degeneracy along $k_y = 0$. With this term there remains a spinless version of time-reversal symmetry $T = \mathcal{K}$. Using the argument in the previous subsection, we obtain that in the weak-pairing limit, such a model hosts gapless hinge modes. We indeed confirmed this by numerically solving the lattice model at a finite system size. Unfortunately, however, in general the correct form of the $T'$-breaking perturbation that fully disentangles the Weyl bands from remote bands depends on the detailed model and requires a case-by-case analysis.

# 7 Conclusion

In this work, we have shown that in a time-reversal symmetric doped Weyl semimetal, the combination of symmetry constraints ($\tilde{R}_{4z}$ and $T$) and momentum space structure of a finite-range attractive interaction naturally leads to a chiral superconducting state. By analyzing the topological properties of the superconducting state, we show identify it is a second-order topological phase with chiral Majorana hinge modes traversing the surface.

We have also analyzed the classification of general BdG Hamiltonians with four-fold rotoinversion symmetry supporting second-order topology and found that the classification to be $\mathbb{Z}_2$. We show that the hinge modes can be removed by inducing strong surface interaction leading to a topologically ordered surface state. Crucially such a topologically ordered system with four-fold rotoinversion symmetry cannot be realized in strictly two dimensions (i.e without a three dimensional bulk) and is therefore anomalous. The less constrained system with only twofold symmetry is shown to be boundary-obstructed while also hosting chiral Majorana hinge modes.

In a broader context, Our work showed that the nontrivial topology and gapless excitations in a topological semimetal provide a natural platform for novel topological superconductivity. It will be interesting to explore possible topological superconducting phases from other types of topological semimetals.

# Acknowledgements

We thank Ming-Hao Li, Titus Neupert and Sid Parameswaran for useful discussions. AT acknowledges funding by the European Union's Horizon 2020 research and innovation program under the Marie Sklodowska Curie grant agreement No 701647. AJ and YW are supported by startup funds at the university of Florida.

# A  A comment on the Wannier spectrum of the model with $\tilde{R}_{4z}$

The Wannier spectrum come form diagonalizing the Wannier Hamiltonian $\hat{\nu}_i(\boldsymbol{k})$ defined through the Wilson loops in the $i$-th direction,

$$e^{i2\pi\hat{\nu}_i(\boldsymbol{k})} \equiv \prod_{n=0}^{L_i-1} \mathcal{P}(\boldsymbol{k} + 2\pi n\boldsymbol{e}_i/L_i), \tag{105}$$

where $L_i$ is the system size along $i$-th direction, and $\mathcal{P}(\boldsymbol{k}) = \sum_{i=1,2} |u_i(\boldsymbol{k})\rangle\langle u_i(\boldsymbol{k})|$ is the projection operator on the occupied states. We note that the operator on the RHS of the above equation acts on a 4-dimensional Hilbert space. However, because of the projection operators involved, it has a 2-dimensional null space, and effectively the Wannier Hamiltonian, $\hat{\nu}_i(\boldsymbol{k})$, is 2-dimensional.

When considering only internal symmetries, the Wannier spectrum in the $i$-th direction share the same topological properties with the surface of the system perpendicular to that direction. [78] However, spatial symmetries can impose vastly different constrains on the surface bands and the Wannier bands, thus leading to different topological features. Indeed for our case, the four-fold rotoinversion symmetry act very differently on the Wilson loop in the $z$-direction and the surface perpendicular to it. The four-fold rotoinversion symmetry maps

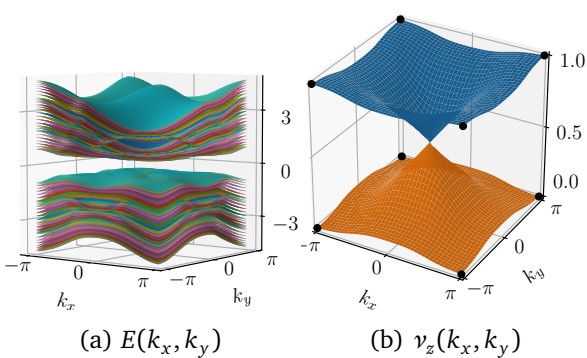

(a) $E(k_x, k_y)$       (b) $\nu_z(k_x, k_y)$

Figure 14: The energy spectrum (a) with open boundary conditions in the $z$-direction. Wannier spectrum (b) for the Wilson loops in the $z$-direction. Both graphs are in the topological phase of the system, $\gamma = 0, \Delta = 0.4, \mu = 0.5$. Size $= (15 \times 15)$. The gaplessness of the Wannier spectrum is protected by $\tilde{R}_{4z}$ while the surface energy spectrum can be gapped without breaking the symmetry.

the top surface of the sample to the bottom surface of the sample, and thus does not put any constrains on the surface spectrum.

Consider the action of the four-fold rotoinversion symmetry on $\mathcal{W}_z(\boldsymbol{k})$ is,

$$\tilde{R}_{4z}\hat{\mathcal{W}}_z(\boldsymbol{k})\tilde{R}_{4z}^{-1} = \hat{\mathcal{W}}_z^{\dagger}(\tilde{R}_{4z}\boldsymbol{k}), \tag{106}$$

which puts the following constraint on the Wannier spectrum,

$$\{\nu_z^i(k_x, k_y)\} = \{-\nu_z^i(k_y, -k_x)\} \quad \text{mod. } 1. \tag{107}$$

This action can be thought of as a combination of a chiral symmetry and a fourfold rotation symmetry. In 2D a chiral symmetry can lead to a symmetry protected Dirac point. We explicitly calculate the Wannier spectrum, and the surface bands for open boundaries in the $z$-direction and compare them. When the chemical potential is zero, we have both spectra to be gapless. However, the gapless mode in the Wannier spectrum is protected by the action of the $R_{4z}$ operator, while gapless mode in the surface spectrum is accidental. Indeed, for non-zero chemical potential, we see that the surface spectrum opens a gap, while the Wannier spectrum does not, see Fig. 14.

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
