# Peer review of "Higher-order topological superconductors from Weyl semimetals"

_SciPost Physics, doi:SciPost Phys. 12, 053 (2022)_

## Round 1 · Referee Report · Frank Schindler (Referee 1) · 2021-6-1

Strengths

1- connects theory to physical realization 2- presents a full, self-contained story 3- is fairly explicit/ comprehensible

Weaknesses

1- some claims are not generic/ require fine-tuning 2- classification results are not presented in context of extensive literature

Report

This manuscript comprehensively addresses a route to higher-order topological superconductors via long-range pairing in doped Weyl semimetals. It describes the normal state Hamiltonian, explicitly derives the BdG mean field theory, analyses its topological properties, and even treats the effects of surface topological order. Because physical realizations of higher-order superconductors are hard to come by, I believe that the paper is in principle worthy of publication. However, there are a couple of issues (detailed below) that need to be addressed before I can make such a recommendation.

Requested changes

1- Why should on-site/ short-range pairing be disallowed? The case for higher-order topology could be argued much more compellingly if a mechanism favoring long-range pairing is given.

2- It should be clearly stated to what extent the classification results of Sec. 4 are original with respect to established references such as Science Advances eaaz8367 and Phys. Rev. Research 3 023086.

3- The Hamiltonian/symmetry discussion in 2.1 needs to connect to physical degrees of freedom. Right now, the Hamiltonian is stated, and then the representation of R4 symmetry is derived from it. This approach has two problems: the physical interpretation of the "internal band space" remains unclear, and R4 symmetry is not guaranteed. That is, the operator in Eq. (8) need not be a good R4 symmetry depending on the momentum-space structure of f(k) away from k=0. To ameliorate both problems, the authors should first state the microscopic orbitals, then derive the symmetry operations (T and R4) from them, and then derive the constraints these symmetries impose on the Hamiltonian.

4- Fig 5 shows eigenvalues that are not compatible with TRS (they do not come in complex-conjugated pairs). This can't be correct, because Eq. (35) is time-reversal symmetric.

5- This claim on page 8 should be derived explicitly: "independent of the details of the band structure, ... the triplet channel ... is always an eigenmode of Eq. (23)."

6- The quantity "Berry flux" should be defined before it is used. It is particularly confusing that the symbol "C" is used, suggesting a Chern number, but there seems to be a mismatch by a factor of 2π.

7- Figure 3 should be referenced and explained.

8- "although here due to the lack of SU (2) symmetry in the band space, the four components are in general mixed": What are the "four components" here?

9- When the case with T^2=-1 is discussed on page 8, the use of R4 eigenvalues \pm 1 is inconsistent. T^2=-1 implies a spinful system where R4 has eigenvalues e^{i n π/4} with n an odd integer.

10- The sentence "A minimal model of such a system consists of two bands with four co-planar Weyl points." contains a hidden assumption (Weyl points not at high-symmetry points) that should be spelled out.

11- In the introduction it should be mentioned that the authors consider spinless electrons/ no SOC so that T^2 = +1.

12- The sentence " a tube enclosing the hinge has a second Chern number protected by R4z symmetry" in the introduction is confusing because an individual hinge breaks R4z symmetry.

13- The sentence "Weyl semimetal with four Weyl nodes with attractive interaction naturally host a higher-order topological superconducting phase" is misleading because the authors show this only for long-range attractive interactions without arguing why these should be relevant in real materials.

Typos: 14- "Nielson-Ninomiya" on page 4 15- "the Green’s function G(k, ω) ≡ G(k, ω) = ..." on page 7 16- Eq. (28) has one minus sign too much 17- "Chiral" is capitalized two times on page 10

  • validity: high
  • significance: good
  • originality: good
  • clarity: top
  • formatting: excellent
  • grammar: excellent

Author:  Ammar Jahin  on 2021-07-26  [id 1614]

(in reply to Report 1 by Frank Schindler on 2021-06-01)

We thank the referee for the useful and insightful comments.

  1. Why should on-site/short-range pairing be disallowed? The case for higher-order topology could be argued much more compellingly if a mechanism favoring long-range pairing is given.

    • We do not argue that short-ranged paring is disallowed in the pairing process. Rather, we show that as long as the pairing interaction is finite-ranged, it naturally induces $p$-wave pairing order with higher-order topology. We emphasize that this pairing mechanism does not necessarily require long-range interaction -- our requirement only excludes e.g. on-site interaction, which is featureless in momentum space. In light of this comment, we have edited the text to make this point clearer.
  2. It should be clearly stated to what extent the classification results of Sec. 4 are original with respect to established references such as Science Advances eaaz8367 and Phys. Rev. Research 3 023086.

    • The classification derived in Sec.4 of our manuscript is applicable to but not limited to single particle/mean field descriptions of higher-order superconductors. In contrast both references pointed out by the referee are formulated within the mean-field BdG formalism. Another notable distinction is that our approach is based on a real space picture wherein we construct equivalence classes of ground state wavefunctions while the approach of the references is based on symmetry indicators constructed from symmetry eigenvalues at high-symmetry points in the Brillouin zone. We have added such a discussion in the edited text.
  3. The Hamiltonian/symmetry discussion in 2.1 needs to connect to physical degrees of freedom. Right now, the Hamiltonian is stated, and then the representation of R4 symmetry is derived from it. This approach has two problems: the physical interpretation of the "internal band space" remains unclear, and R4 symmetry is not guaranteed. That is, the operator in Eq. (8) need not be a good R4 symmetry depending on the momentum-space structure of f(k) away from k=0. To ameliorate both problems, the authors should first state the microscopic orbitals, then derive the symmetry operations (T and R4) from them, and then derive the constraints these symmetries impose on the Hamiltonian.

    • We thank the referee for this suggestion. We believe this is a matter of taste. To reiterate our strategy, we consider a generic ''minimal model" for a time-reversal Weyl semimetal with $\mathsf R_{4z}$ symmetry, and we show that remarkably the form of symmetry operators can be completely fixed. The symmetry action on Wannier orbitals forming the Weyl semimetal can then be seen from the symmetry operators, such as that in Eq. (8). In our opinion, if we were constructing an effective Hamiltonian for a realistic material, the approach suggested by the Referee would have been more suitable. However in this work we analyze a generic model without a particular material in mind, we believe the approach we take is adequate.
  4. Fig 5 shows eigenvalues that are not compatible with TRS (they do not come in complex-conjugated pairs). This can't be correct, because Eq. (35) is time-reversal symmetric.

    • We believe this is a simple misunderstanding. The time-reversal symmetry squares to $1$ and thus it does not impose Kramer degeneracy. Indeed the eigenvectors at these high-symmetry points are invariant under the action of time-reversal symmetry.
  5. This claim on page 8 should be derived explicitly: "independent of the details of the band structure, ... the triplet channel ... is always an eigenmode of Eq. (23)."

    • This derivation is in Appendix A. In light of the potential interest to the readers, in the new manuscript we have moved it into the main text.
  6. The quantity "Berry flux" should be defined before it is used. It is particularly confusing that the symbol "C" is used, suggesting a Chern number, but there seems to be a mismatch by a factor of $2\pi$.

    • We thank the referee for pointing this out and we have fixed it.
  7. Figure 3 should be referenced and explained.

    • We thank the referee for pointing this out and we have properly referenced and discussed Figure 3 in the revised version.
  8. "although here due to the lack of SU (2) symmetry in the band space, the four components are in general mixed": What are the "four components" here?

    • The four components referred to here are those of the paring order parameter, namely $(\Delta_ 0, d_ x, d_ y, d_z)$. We have modified the text to make this point clearer.
  9. When the case with $T^2=-1$ is discussed on page 8, the use of R4 eigenvalues $\pm 1$ is inconsistent. $T^2=-1$ implies a spinful system where R4 has eigenvalues $e^{i n \pi/4}$ with n an odd integer.

    • We thank the refereeing for pointing this out. In the old version the argument against $\mathsf T^2=-1$ was made assuming the form of the $\mathsf R_ {4z}$ operator for the previous $\mathsf T^2=1$ case. This is clearly confusing. In the revised version, we adopted a more general argument showing $\mathsf T^2=-1$ is inconsistent with the $\mathsf R_ {4z}$ symmetry in the Weyl semimetal, which does not rely on the specific form of the $\mathsf R_{4z}$ operator and its eigenvalues.
  10. The sentence "A minimal model of such a system consists of two bands with four co-planar Weyl points." contains a hidden assumption (Weyl points not at high-symmetry points) that should be spelled out.

    • We agree, and have made this change in the revised version.
  11. In the introduction it should be mentioned that the authors consider spinless electrons/ no SOC so that $T^2 = +1$.

    • Indeed, even if we did briefly consider the $\mathsf T^2=-1$ case in Sec. 6.3, our main results are universally applicable to $\mathsf T^2=1$ cases only. We have followed the Referee's suggestion and made the change.
  12. The sentence "a tube enclosing the hinge has a second Chern number protected by R4z symmetry" in the introduction is confusing because an individual hinge breaks R4z symmetry.

    • We thank the referee for raising this issue. The key point here is that the quantization of the defect topological invariant does not require spatial symmetries. Rather, spatial symmetries ensure the topological invariant is quantized to a nontrivial value. To be specific, although the tube does not preserve $\mathsf R_{4z}$ symmetry, as correctly pointed out by the referee, different points on the tube are related to one another by the action of the symmetry. Therefore choosing the mass to have a certain value (i.e. the mass vector to point in a certain arbitrary but fixed direction), automatically pins the value of the mass at certain other points on the tube. Furthermore the value of the masses deep in the bulk as well as outside are also fixed by symmetry requirements. Together these two considerations, pin the topological winding number of the mass texture on the tube. The winding number being directly related to the second Chern number consequently also enforces it, in this case to be quantized as an odd integer.
  13. The sentence "Weyl semimetal with four Weyl nodes with attractive interaction naturally host a higher-order topological superconducting phase" is misleading because the authors show this only for long-range attractive interactions without arguing why these should be relevant in real materials.

    • As we mentioned, our theory does not require long-range interaction. However, we think that the referee's point is a valid point, and we have changed the sentence to "Weyl semimetal with four Weyl nodes with attractive interaction is a promising candidate to host a higher-order topological superconudcting phase."

---

## Round 1 · Referee Report · Rui-Xing Zhang (Referee 2) · 2021-6-6

Strengths

  1. The emergence of intrinsic higher-order topological superconductivity from doped Weyl semimetal is an interesting idea/development in the field.
  2. The results are rather comprehensive and the analysis is professional.
  3. The paper is well-written.

Weaknesses

  1. The applicability of this theory to non-minimal Weyl systems and real-world materials is unclear, possibly limiting the impact of this work.
  2. Some statements are not well justified and could be overclaimed.
  3. Missing some important references.

Report

In this manuscript, the authors introduced an interesting theory for emergent higher-order topological superconductivity (HOTSC) in doped R4z-symmetric Weyl semimetal (WSM). Starting from a minimal model for R4z-compatible WSM, the authors demonstrated that the most favorable pairing channel here is a nodeless odd-parity one, directly leading to an R4z-protected HOTSC phase as a natural outcome. The authors also discussed a possible anomalous surface topological order that can symmetrically eliminate the R4z-protected higher-order topology in a symmetric & adiabatic way. When breaking R4z to C2, possible boundary obstructed extrinsic HOTSC phase is also discussed. As far as I can tell, the paper is well-written and the results in this work are interesting, professional, and comprehensive. However, the applicability of this theory to non-minimal Weyl systems and real-world materials is not quite clear to me. Besides, I also find some statements not very well justified and missing a highly relevant reference. Therefore, I can consider to recommend this work for publication in SciPost after the authors address following questions/comments.

  1. It seems to me that chiral p-wave being the most favorable pairing is a direct result of the dimension of the Hamiltonian matrix. Namely, a four-band Dirac model (including the BdG partners) apparently has much fewer choices of symmetry-allowed mass terms than that of an eight-band model. This is probably why this specific pairing term is the only one that makes the SC state fully gapped, since the kinetic terms and the pairing terms can form a complete set of anticommuting 4 by 4 gamma matrices, as shown in Eq. 27. However, if the model has eight bands instead, I can imagine other pairing terms would be equally favorable in energy for being nodeless.

This observation directly makes me concerned about the applicability of the present theory to a WSM with more than four Weyl nodes. Can a WSM with 4n Weyl points (n>=1) always be described by a two-band model? If yes or no, is the chiral p-wave pairing still the leading instability? Can the same pairing always give rise to intrinsic HOTSC despite the number of Weyl points?

  1. To further enhance the impact of this work, I believe a discussion on possible material candidates will be very helpful. Nevertheless, a basic requirement for the R4z WSM is its spinless nature. I wonder if the authors have any Weyl material candidate in mind that (i) is effective spinless; (ii) preserves some “effective” TRS. If not, can the authors discuss where and how to fulfill the above conditions simultaneously in an actual physical system?

  2. It is surprising that this work has missed an important relevant work on rotoinversion protected HOTI (Phys. Rev. B 98, 081110(R) by van Miert and Ortex). The authors might need to compare the current results with the nonsuperconducting counterpart in this PRB paper, especially a Z2 classification is also found in the HOTI case.

  3. I am a little confused about the definition of the surface Chern number in Eq. 33. In particular, the authors defined “S^(+/-) is the set of upper/lower half of the layers”. Does this mean that for a N-layer slab, we label the layer collection of n=1,…,N/2 to be S^+ and that of n=N/2+1,…,N to be S^-? If so, I don’t quite get the relation in Eq. 34 as well. For example, I would expect C_yz^+ = C_yz^-=-0.5 due to the C2 symmetry. To make the hinge modes on the bottom xy surface to appear, I would require C_xy^- = +0.5 = -C_xy^+ to achieve the desired quantized change of surface Chern number. Am I correct?

  4. Two comments on the Wannier obstruction discussed in Section 3.2.

(i) It has been known that being Wannier obstructed is a neither necessary nor sufficient condition for being higher-order (or first-order) topological for BdG systems (see examples in Ref. 35 and 54). This is crucially different from the story in non-SC electronic insulators. Therefore, as far as I am concerned, proving the existence of a Wannier obstruction in a BdG system does NOT equal to the proof of higher-order topology, and vice versa. So the authors should be careful about making relevant claims. For example, the statement in Sec. 3.2 “…generally we have proven that an R4z Weyl semimetal with four Weyl nodes with attractive interaction naturally host a higher-order topological superconducting phase…” is NOT quite accurate to me, since only a proof of Wannier obstruction is shown in that section.

To justify the above statement, the authors need to complete a proof or at least an argument of bulk-boundary correspondence (BBC) by showing that why such a Wannier obstruction will necessarily lead to the hinge modes. If such a proof of BBC is absent or difficult, I would suggest the authors modify the above claim accordingly.

(ii) Can the authors quickly check if the Wannier obstruction is stable or fragile?

  1. Related to the above question of BBC, the definition of the defect-based second Chern number looks very interesting. Is it possible to relate this boundary-related invariant o the previous bulk symmetry indicator for Wannier obstruction? If this can be done, it might help explain the BBC.

Some minor comments:

  1. The basis of V_II’ matrix in Eq. 17 is not clear to me. It would be better to relate the notation of I and I’ to that in Fig. 1.

  2. Slightly above Eq. 20, there is a typo in the definition of the Green function.

  3. I find the Wannier spectrum calculation in Fig. 13 (b) of Appendix B quite useful to understand the bulk Wannier obstruction. So I would suggest move this figure to Section 3.2.

  4. About the definition of the second Chern number in 3.3.2, I would suggest visualizing the choice of hinge, S_\gamma^1, and other geometric quantities in a new schematic figure, which will definitely help explain the geometric meaning of the topological invariant.

  5. At the beginning of Sec. 4, a definition for n-cell is missing and this concept might not be familiar to general readers.

  • validity: good
  • significance: good
  • originality: good
  • clarity: good
  • formatting: excellent
  • grammar: excellent

Author:  Ammar Jahin  on 2021-07-26  [id 1615]

(in reply to Report 2 by Rui-Xing Zhang on 2021-06-06)

We thank the referee for the useful and insightful comments.

  1. It seems to me that chiral p-wave being the most favorable pairing is a direct result of the dimension of the Hamiltonian matrix. Namely, a four-band Dirac model (including the BdG partners) apparently has much fewer choices of symmetry-allowed mass terms than that of an eight-band model. This is probably why this specific pairing term is the only one that makes the SC state fully gapped, since the kinetic terms and the pairing terms can form a complete set of anticommuting 4 by 4 gamma matrices, as shown in Eq. 27. However, if the model has eight bands instead, I can imagine other pairing terms would be equally favorable in energy for being nodeless. This observation directly makes me concerned about the applicability of the present theory to a WSM with more than four Weyl nodes. Can a WSM with 4n Weyl points ($n>=1$) always be described by a two-band model? If yes or no, is the chiral p-wave pairing still the leading instability? Can the same pairing always give rise to intrinsic HOTSC despite the number of Weyl points?

    • This is an interesting question. While we focused on a minimal model for a Weyl semimetal with $\mathsf R_{4z}$ and time-reversal symmetries, our results can be straightforwardly generalized to the case with more Weyl points. As the referee hinted, there are two routes toward more Weyl points -- either by including additional copies of the two-band model, or by modifying the dispersion of the two-band model (for example via including longer-range hopping terms).
      • In the first case, the generalization is straightforward -- since the classification we obtained is $Z_2$, then the chiral superconducting system with $4n$ Weyl points has a nontrivial topology if $n$ is odd.
      • In the second case, one can repeat the same analysis for the modified two-band normal state, via the symmetry indicator approach or via the defect approach. It is not difficult to show that, again, the higher-order topology is nontrivial if $n$ is odd. We thank the referee for raising this point and we have added a discussion in the revised text, which we believe improves the quality of our work.
  2. To further enhance the impact of this work, I believe a discussion on possible material candidates will be very helpful. Nevertheless, a basic requirement for the R4z WSM is its spinless nature. I wonder if the authors have any Weyl material candidate in mind that (i) is effective spinless; (ii) preserves some “effective” TRS. If not, can the authors discuss where and how to fulfill the above conditions simultaneously in an actual physical system?

    • An effective ''spinless" TRS can be realized by combining the physical TRS with certain spin rotation symmetries. Unfortunately, we are not immediately aware of a material realization. We are hopeful that with the recent development of topological quantum chemistry based on symmetry indicators, such candidate Weyl materials with $\mathsf{R} _{4z}$ symmetry can soon be identified. However, we believe this is out of the scope of the current work, and postpone such an investigation to future studies.
  3. It is surprising that this work has missed an important relevant work on rotoinversion protected HOTI (Phys. Rev. B 98, 081110(R) by van Miert and Ortex). The authors might need to compare the current results with the nonsuperconducting counterpart in this PRB paper, especially a Z2 classification is also found in the HOTI case.

    • We thank the referee for pointing out this important reference. Although our methodology is very different, our results do agree with those in the reference. An important difference between our methodology and that of the work by van Miert and Ortex is that our approach is applicable to interacting as well as discordered/non-translational invariant systems that preseve the rotoinversion symmetry. Although we are interested in higher-order superconductors in the present work, our classification scheme can be readily adapted to insulators as well, by simply modifying the 2D blocks one uses in the cell decomposition described in Sec.4 of our paper. More precisely, the 2D blocks which host the $p+ip$ phase in the superconducting case, host Chern insulator phases in the insulating case. Upon doing so, we recover the $Z_2$ classification of van Miert and Ortex.
  4. I am a little confused about the definition of the surface Chern number in Eq. 33. In particular, the authors defined “S^{+/-} is the set of upper/lower half of the layers”. Does this mean that for a N-layer slab, we label the layer collection of n=1,…,N/2 to be S^+ and that of n=N/2+1,…,N to be S^-? If so, I don’t quite get the relation in Eq. 34 as well. For example, I would expect C^+ {yz} = C^- {yz} =-0.5 due to the C2 symmetry. To make the hinge modes on the bottom xy surface to appear, I would require C^- {xy} = +0.5 = -C^+ {xy} to achieve the desired quantized change of surface Chern number. Am I correct?

    • Yes, this mentioned definition of $S^{+/-}$ is correct. We believe this was confusing because in the old version we had the direction normal to the surfaces to be fixed relative to the $x, y, z$-directions. We now adopted a more natural way of defining the normal vectors to be pointing outside of the sample. With such choice, everything is consistent with what is mentioned above. We thank the referee for point this out.
  5. Two comments on the Wannier obstruction discussed in Section 3.2.

    1. It has been known that being Wannier obstructed is a neither necessary nor sufficient condition for being higher-order (or first-order) topological for BdG systems (see examples in Ref. 35 and 54). This is crucially different from the story in non-SC electronic insulators. Therefore, as far as I am concerned, proving the existence of a Wannier obstruction in a BdG system does NOT equal to the proof of higher-order topology, and vice versa. So the authors should be careful about making relevant claims. For example, the statement in Sec. 3.2 “…generally we have proven that an R4z Weyl semimetal with four Weyl nodes with attractive interaction naturally host a higher-order topological superconducting phase…” is NOT quite accurate to me, since only a proof of Wannier obstruction is shown in that section. To justify the above statement, the authors need to complete a proof or at least an argument of bulk-boundary correspondence (BBC) by showing that why such a Wannier obstruction will necessarily lead to the hinge modes. If such a proof of BBC is absent or difficult, I would suggest the authors modify the above claim accordingly.
      • We thank the referee for pointing this out. Indeed, in general, Wannier obstruction in the bulk is neither a sufficient nor a necessary condition for hinge modes. However, for our case, it is not difficult to provide a direct connection between the bulk obstruction and the chiral hinge modes. To this end, we note that both of the two-dimensional subsystems at $k_ z=0$ and $k_ z=\pi$ are Wannier representable. However, the Wyckoff centers of the Wannier states for the two subsystem are different -- it is at an obstructed position $(1/2,1/2)$ for $k_ z=0$ and at $(0,0)$ for $k_ z=\pi$. (For this reason there is no compatible Wannier representation for the full 3d system.) From our previous work (PRR, 043300 (2020)), this means that there is a corner zero mode state at $k_ z=0$, which disperses away at $k_ z =\pi$. Such a spectral flow at the hinge directly corresponds to the chiral hinge states. Inspired by the referee's comment, we have added such a discussion into the revised text.
    2. Can the authors quickly check if the Wannier obstruction is stable or fragile?
      • From our previous answer directly connecting the Wannier obstruction with the chiral hinge modes, it is stable.
  6. Related to the above question of BBC, the definition of the defect-based second Chern number looks very interesting. Is it possible to relate this boundary-related invariant o the previous bulk symmetry indicator for Wannier obstruction? If this can be done, it might help explain the BBC.

    • We are not aware of any direct connection, although clearly they are obtained from the same set of information -- the pattern of the mass terms at high-symmetry points. Regarding the bulk-boundary correspondence, we believe our previous answer has clarified this point.
  7. The basis of $VII’$ matrix in Eq. 17 is not clear to me. It would be better to relate the notation of I and I’ to that in Fig. 1.

    • We thank the referee for bringing this up to our attention. We have modifed the main text and hope it is clearer now.
  8. Slightly above Eq. 20, there is a typo in the definition of the Green function.

    • We thank the referee for pointing out this typo.
  9. I find the Wannier spectrum calculation in Fig. 13 (b) of Appendix B (now Appendix A in the new version) quite useful to understand the bulk Wannier obstruction. So I would suggest move this figure to Section 3.2.

    • We thank the referee for the interest. Appendix B and Fig. 13 (b) address a subtle difference between the Wannier spectrum and the surface energy spectrum in the presence of $\mathsf R_ {4z}$ symmetry -- even if the $xy$ surface is generally gapped, the Wannier spectrum in $(k_{x},k_{y})$ is gapless, reflecting the bulk Wannier obstruction. Heeding the referee's suggestion, we have made a reference to Fig. 13 (b) in the main text. However, the discussion on Fig. 13 is rather disconnected from the the main text. We hope our response above and the edited manuscript have made the bulk Wannier obstruction clear, and prefer to keep this figure in the Appendix.
  10. About the definition of the second Chern number in 3.3.2, I would suggest visualizing the choice of hinge, $S_\gamma^1$, and other geometric quantities in a new schematic figure, which will definitely help explain the geometric meaning of the topological invariant.

    • We thank the referee for bringing this up to out attention. We have included such figure in the new version of the text.
  11. At the beginning of Sec. 4, a definition for n-cell is missing and this concept might not be familiar to general readers.

    • We thank the referee for bringing this up to out attention. We have modified this in the new version of the text.

---

## Round 2 · Referee Report · Frank Schindler · 2021-8-3

Report

In their resubmission, the authors have addressed most of my requests and comments in a satisfactory manner. There is only one remaining issue:

In my first report, I had noted that:
"Fig 5 shows eigenvalues that are not compatible with TRS (they do not come in complex-conjugated pairs). This can't be correct, because Eq. (35) is time-reversal symmetric."

The authors replied with:
"We believe this is a simple misunderstanding. The time-reversal symmetry squares to 1 and thus it does not impose Kramer degeneracy. Indeed the eigenvectors at these high-symmetry points are invariant under the action of time-reversal symmetry."

This reply does not address my concern. Even when time-reversal symmetry squares to +1, so that there is no Kramers degeneracy, it still enforces a pairing of complex R4z eigenvalues. For this, we assume an eigenstate R4z |Ψ> = λ |Ψ>. Then, we find R4z T |Ψ> = T R4z |Ψ> = λ* T |Ψ>.

Given that Fig. 5 shows R4z eigenvalues that are not complex-conjugate to one another or real, the assumption that the commutator [T, R4z]=0 vanishes must be violated. This is unphysical -- all spatial symmetries should commute with time-reversal -- and needs to be amended before I can recommend publication of the manuscript.

  • validity: -
  • significance: -
  • originality: -
  • clarity: -
  • formatting: -
  • grammar: -

Author:  Ammar Jahin  on 2021-09-02

(in reply to Report 1 by Frank Schindler on 2021-08-03)

We thank the referee for the further clarification, and indeed our previous response was not completely satisfactory.

The key question is that whether one should necessarily require $[T, R_{4z}]=0$. To this end, we would like first invoke a familiar conclusion — a two-fold symmetry necessarily commutes or anti-commutes with time-reversal symmetry. Indeed, in several well-known works on classifying topological crystalline insulators (TCI), e.g., the work by Shiozaki and Sato, the classification of a TCI with mirror symmetry M depends on whether $MT= TM$ or $ MT=-TM$.

In fact, here $MT=\pm TM$ are the only two options. This is due to the fact that $M, T$ and$ MT$ are two-fold symmetries. Since $MT$ is an anti-unitary symmetry, we can write $MT = U K$, where $U$ is a unitary operator and $K$ is complex conjugation. Then we have $(MT)^2 = UU^* = U(U^{-1})^T = \phi$. Since $MT$ is a two-fold symmetry, $\phi$ must be a diagonal operator. Therefore, $U = \phi U^T \phi$. Explicitly expressing the components, this is only possible if $\phi=\pm 1$. We thus have $MTMT=\pm 1$. Further assuming $M^2 =1$, and given that $T^2 = \pm 1$, we have $MT = \pm TM$.

However, to the best of our knowledge, no such constraints exist for a generic symmetry, including our four-fold symmetry $R_{4z}$. A counterexample is a generic spin rotation symmetry $S = \exp(i\sigma_z\theta/2)$, and a spinless time-reversal symmetry $T = K$. For a generic angle $\theta$, $ S$ and $T$ neither commute nor anti-commute. But this scenario is absolutely physical.

Going back to our case, time-reversal symmetry is broken in general but is restored at high-symmetry points. First, as we mentioned, the two $R_{4z}$ eigenvalues at every $K$ point in Fig. 5 are not TR partners. Instead each of them is invariant under TR. Second, because $R_{4z}$ does not commute or anti commute with T, its eigenvalues does not need to be real. In fact, our particular form of $R_{4z}$ in the BdG Hamiltonian comes from the fact that the $p+ip$ pairing order parameter carries an orbital angular momentum 1. In order to maintain rotation symmetry, a rotation in the pseudospin degree of freedom $\vec\tau=(\tau_x,\tau_y)$ needs to be incorporated into the rotation operator, i.e., $R_{4z} \propto \exp(i\tau_z \pi/4)$. This makes our case quite similar to the example with S above.

It is instructive to look at $R_{4z}^2=C_2$, which is a twofold symmetry. Its eigenstate $C_2 |Ψ\rangle = λ |Ψ\rangle$ satisfies the following relation: $ C_2 T |Ψ\rangle = \pm T C_2 |Ψ\rangle = \pm λ^* T |Ψ\rangle,$ where the $\pm$ depends on whether $C_2$ and $T$ commute or anti-commute. We see that if $T |Ψ\rangle = |Ψ\rangle$, then $λ$ needs to be either real (if $C_2$ and $T$ commute) or imaginary (if they anti-commute). In our case, $C_2 = i\tau_x$ and $T=K$, and we have $C_2 T = -T C_2$. Therefore $C_2$ eigenvalues are required to be imaginary. Indeed, by squaring the eigenvalues in Fig. 5, all $C_2$ eigenvalues are imaginary, consistent with the requirement of time-reversal symmetry.

We hope this addresses the referee’s insightful question.

---

## Round 2 · Referee Report · Rui-Xing Zhang · 2021-8-25

Report

In both the resubmitted manuscript and the reply letter, the authors have done an excellent job in addressing the questions and concerns that I had in my previous report. In particular, they have significantly improved some figures and notations to greatly enhanced the readability of this work. Besides, they have clearly clarified the connection with previous works. Even though it remains unclear what real-world material system can serve as a platform to realize the proposed physics, we generally do not expect the first theory proposal to resolve all the issues that one might encounter. Now I think the comprehensiveness and novelty of this paper make it a beautiful theory work and SciPost-worthy on its own, and the material search will be a next-level future problem. Therefore, I am happy to recommend the current manuscript for publication in SciPost.

---

## Round 2 · Author Response

The authors would like to thank the referees for their insightful comments that helped this work take a better shape.

---

## Round 2 · List of Changes

# List of Changes

- Move the discussion of appendix A to the main text
- Add definition of the $n$-cells.
- Add some discussion about how to generalize to more than $4$ Weyl points.
- Change the directions of the vector normal to the surfaces in Eq. (65).
- Add the discussion about how the Wannier obstruction directly leads to the chiral hinge mode.
- Make point about the requirement for long range interaction more clear.
- Add comparisons to previously found results.
- Modify the notation for $V_ {II^\prime}$.
- Fixed the factor of $2\pi$ between Berry flux and Chern number.
- Add what are the four components are under Eq. (20b).
- Properly reference Fig. 3 in the main text.
- Change the discussion about why $\mathsf T^2 = -1$ is inconsistent with our model.
- Mention that we take the Weyl points not to sit at the high-symmetry points.
- Change the introduction to include that we mainly consider spinless fermions.
- Fix typo above above Eq. (22) in the definition of the Green's function.
- Include a figure for the choice of the path $S^1_{\gamma}$.

---

## Round 3 · Referee Report · Frank Schindler · 2021-11-15

Report
After carefully considering the author's response, I remain unconvinced that my concern is satisfactorily addressed.
Let me recall that I had stated previously: "Given that Fig. 5 shows R4z eigenvalues that are not complex-conjugate to one another or real, the assumption that the commutator [T, R4z]=0 vanishes must be violated. This is unphysical -- all spatial symmetries should commute with time reversal -- and needs to be amended before I can recommend publication of the manuscript."
This inconsistency is still present in the current version of the manuscript.
In their reply, the authors note that in certain works in the classification literature, spatial symmetries are allowed to anti-commute rather than commute with time-reversal symmetry. This is a mathematical trick: If a spatial symmetry R anti-commutes with T, then i*R commutes with T due to the anti-unitary nature of time reversal. In this case, i*R is the physical symmetry.
Moreover, the authors allege that another counterexample to my statement is given by spin rotation symmetry $S = e^{i \sigma_z \theta/2}$, and spinless time-reversal symmetry $T=K$. This is in fact no counterexample, but merely an inconsistent choice of symmetries. For spinful symmetries, spinful time-reversal $T = i \sigma_y K$ should be used, which clearly commutes with spin rotations.
I therefore maintain my previous concern and cannot recommend publication of the manuscript in its present form.
Author: Ammar Jahin on 2021-12-07 [id 2015]
(in reply to Report 1 by Frank Schindler on 2021-11-15)In the latest reports, Referee Dr.~Frank Schindler reiterated that from a physical point of view, the roto-inversion operator $\mathsf{R}_{4z}$ and time-reversal $\mathsf{T}$ must commute. According to the referee, while in certain examples, the commutation between the spatial symmetry operator and time-reversal operator may be violated, it can be restored if the operators are chosen to their ``physical" forms. Referee Prof.~Titus Neupert very nicely summarized that the disagreement stemmed from two different conventions of defining symmetry operators, and asked us to elucidate the symmetry operations from the perspective of electronic orbitals.
We fully agree with both referees' point of view, and we thank them for helping us better understand their concerns. It turns out our initial result is fully consistent with the symmetry representation of orbital degrees of freedom, which can be clarified by a modification of notation.
Interpretation of roto-inversion symmetry in the normal state
Let us first consider the normal state on which time-reversal is represented as $\mathsf T = K$, and and rotoinversion (which is a combination of $C_ {4z}$ and $M_ z$) operator is given by $\mathsf R_ {4z} = \hat f_ 1(0) \sigma_ x + \hat f_ 3(0) \sigma_ z$, where $\sigma$ represents our psudespin degree of freedom, and $\hat f^2_ 1(0) + \hat f^2_ 3(0) = 1$, as given in Eq. (8) in our paper. Importantly, here $\mathsf R_ {4z}$ and $ \mathsf{T}$ do commute.
Such a choice of symmetry operators can indeed be understood from the perspective of orbital degrees of freedom. Without loss of generality, let us consider the special case of $\hat f_ 1(0) = 0$, and $\hat f_ 3(0) = 1$, such that $\mathsf R_ {4z} = \sigma_ z$. In this case, we have two orbitals per unit cell both located at rotoinversion invariant points. The $|\sigma_ z = 1\rangle$ orbital can be chosen to be , e.g., an $s$-orbital, while the $|\sigma_ z = -1\rangle$ can be a $d_ {xy}$-orbital. This choice is not unique; for example $|\sigma_ z = 1\rangle=|d_ z^2\rangle$ and $|\sigma_ z = - 1\rangle = |p_ z\rangle$ would also work.
For general values of $\hat f_ 1(0)$ and $\hat f_ 3(0)$, the physical interpretation of $|\sigma_ z = \pm 1\rangle$ orbitals can be obtained as a linear superposition of $|s\rangle$ and $|d_ {xy}\rangle$ via a unitary transformation. Generally, the $\mathsf R_{4z} $ degree of freedom can be realized by, e.g., a (spin-polarized) $s$-orbital or a $d_ {z^2}$-orbital, while the $|\mathsf R_ {4z} =-1\rangle$ degree of freedom can be realized by, e.g. a $d_ {xy}$-orbital or a $p_ z$-orbital.
Non-commutation between roto-inversion symmetry and time-reversal symmetry in the BdG Hamiltonian
Now we move on to the superconducting state with the $p+ip$ pairing term $H_ {\Delta} = \int_ {k} \Delta^{\alpha \beta} (k) c^\dagger_ {\alpha}(k) c^\dagger_ {\beta}(- k)$, which transforms under rotoinversion in the non-trivial one-dimensional representation as $\mathsf R_ {4z} : H_ {\Delta} \mapsto - i H_ {\Delta} $, since it carries angular momentum $1$. Consequently, the mean field Hamiltonian $H = H_ n + H_ {\Delta} + H^\dagger_ {\Delta}$, is not invariant under the action of operator $\mathsf R_ {4z} = \hat f_ 1(0) \sigma_ x + \hat f_ 3(0) \sigma_ z$.
We note that the additional factors of $i$ in front of the pairing terms that spoil the symmetry can be removed by an additional $U(1)$ transformation: $U(\frac{\pi}{4}) : c^\dagger_a (k) \rightarrow e^{i\frac{\pi}{4}} c^\dagger_a (k)$, and $U(\frac{\pi}{4}) : c_a (k) \rightarrow e^{-i\frac{\pi}{4}} c_a (k)$. This $U(1)$ operation leaves the normal part of the Hamiltonian unchanged. Therefore, we define a new operator $\tilde R_ {4z} = U(\frac{\pi}{4}) \mathsf R_ {4z} $. (In the paper we had an abuse of notation in which we did not give this new combined symmetry a new symbol. We admit this is confusing, and have now fixed this.) We now have $[\tilde R_ {4z}, \mathsf T] \neq 0$ because of the added $U(1)$ factor. Crucially, even if $\tilde R_ {4z}$ is not the ``physical" roto-inversion operator, its action on the spatial part of the Hamiltonian is identical to roto-inversion. Indeed, we relied on the eigenvalues of $\tilde R_ {4z}$ to find the Wannier representation of the BdG Hamiltonian by treating it as that of an insulator.
In the Nambu space, this $U(1)$ operator is represented by $e^{-i\frac{\pi}{4} \tau_ z}$, and thus we have $\tilde R_ {4z} = e^{i\frac{\pi}{4} \tau_ z} \mathsf R_ {4z} = e^{-i\frac{\pi}{4} \tau_ z} (\hat f_ 1(0) \sigma_ x + \hat f_ 3(0) \sigma_ z) $, as given in the paper. We note that constructing the symmetry operator for the BdG Hamiltonian starting form the symmetry operator in the normal state and the one-dimensional representation of the pairing terms in the way we just described follows closely the constuction given in Eqs. (7), (8), and (9) of [Trifunovic 2019] (https://arxiv.org/pdf/1910.11271.pdf).
We also note that this additional unitary operator in the Nambu space plays a similar role as $M_ y$ does in the example given by Referee Titus Neupert in obtaining $[M_ x,\tilde T]\neq 0$, although there the modification was on the time-reversal operator $\tilde T \equiv M_ y T$.
Finally, we also note that our choice of time-reversal symmetry $\mathsf{T}=K$ is, strictly speaking, also not ``physical", since the physical time-reversal symmetry operator squares to $-1$. As is well understood and similar to what was mentioned by Referee Neupert, here $\mathsf{T}=K$ can be understood as a composite of the physical time-reversal symmetry $\mathsf{T}' = -is_ y K$ and a spin rotation $is_y$. In this context, we assume the band electrons we consider is fully spin polarized.

---

## Round 3 · Referee Report · Titus Neupert · 2021-11-16

Report
I have been asked by the editor to comment specifically on the issue of symmetry representations that has not been settled between authors and referee so far. I understand that there are two schools of thought about how to define symmetries in such tight-binding models. One strives to derive the symmetry representations from an orbital realization of a model (as the referee advocates), the other one simply posits symmetry representations (as the authors advocate). However, bridging between these two views is important and can be insightful.
Let me do it for the example of mirror symmetry brought up by the authors: A concrete realization could be a spinful chain of s orbitals that is ferromagnetically ordered and spin-orbit coupled. The chain extends in x direction and the magnetization is also along x. Fundamental TRS $T$ ($T^2=-1$) is broken and so is the mirror $M_y$ that sends $y\to -y$ (the latter is broken due to the action of the mirror symmetry $\mathrm{i}\sigma_y$ on the polarized spins). However, the product $\tilde{T}=M_yT$ is a symmetry and along the 1D extent of the system it acts local in real space. Further $\tilde{T}^2=+1$, and $\tilde{T}M_x=-M_x\tilde{T}$, where $M_x$ is the mirror symmetry along the chain. Hence, this is a situation in which the mirror $M_x$ and TRS (meaning the effective TRS $\tilde{T}$, i.e., some local in space antiunitary symmetry) anticommute instead of commute.
I think it would be helpful in connecting the two viewpoints mentioned above, if the authors could produce a similar physical motivation for the choice of symmetry representations that are used in their model, starting from spinful electrons in orbitals and the fundamental action of symmetries in the Lorentz group on Dirac electrons. After all, the authors are not deriving some more abstract problem like a classification table in which all sorts of representations may be listed, but they study a concrete model Hamiltonian with is meant to represent a physical system. To elucidate what type of microscopic system that is, I find such an analysis important and less a matter of taste.
Since TRS should be local and the system is 3D, I guess that arriving at it by a combination of a spatial symmetry with the fundamental TRS, as in my example above, is not an option. Probably assuming a collinear (anti)ferromagnet with negligible SOC, and combining the fundamental TRS with the spin-operator along that conserved spin direction, is more fruitful. If the authors encounter a fundamental obstacle in deriving the symmetry representation in this way, I think it would still be useful for the reader to explain why such a representation is not possible.

---

## Round 3 · List of Changes

* Added more references.
* We addressed the issue raised in report 1 on the second version of the manuscript in a comment to the referee. We believe that no further changes are needed on that front.

---

## Round 4 · Referee Report · Frank Schindler (Referee 1) · 2021-12-13

Report

With their most recent reply and changes to the manuscript, the authors have resolved my remaining concern. The manuscript is now ready for publication.

---

## Round 4 · Referee Report · Titus Neupert (Referee 3) · 2021-12-15

Report

I am very happy to read the clarifying reply of the authors to my report. I have not further objections and recommend publication of the article as is.

---

## Round 4 · List of Changes

• Change notation for the four-fold rotoinversion operator on the BdG level, to avoid confusion as explained in our reply on version 3 of the paper.
  • Include a physical interpretation in terms of band orbitals for the symmetry representation of the normal state given in Eq. (8).

---

## Editorial Decision

published